# Variational Bayes under Model Misspecification

**Yixin Wang**
Columbia University

**David M. Blei**
Columbia University

## Abstract

Variational Bayes (VB) is a scalable alternative to Markov chain Monte Carlo (MCMC) for Bayesian posterior inference. Though popular, VB comes with few theoretical guarantees, most of which focus on well-specified models. However, models are rarely well-specified in practice. In this work, we study VB under model misspecification. We prove the VB posterior is asymptotically normal and centers at the value that *minimizes* the Kullback-Leibler (KL) divergence to the true data-generating distribution. Moreover, the VB posterior mean centers at the same value and is also asymptotically normal. These results generalize the variational Bernstein–von Mises theorem [31] to misspecified models. As a consequence of these results, we find that the model misspecification error *dominates* the variational approximation error in VB posterior predictive distributions. It explains the widely observed phenomenon that VB achieves comparable predictive accuracy with MCMC even though VB uses an approximating family. As illustrations, we study VB under three forms of model misspecification, ranging from model over-/under-dispersion to latent dimensionality misspecification. We conduct two simulation studies that demonstrate the theoretical results.

## 1 Introduction

Bayesian modeling uses posterior inference to discover patterns in data. Begin by positing a probabilistic model that describes the generative process; it is a joint distribution of latent variables and the data. The goal is to infer the posterior, the conditional distribution of the latent variables given the data. The inferred posterior reveals hidden patterns of the data and helps form predictions about new data.

For many models, however, the posterior is computationally difficult—it involves a marginal probability that takes the form of an integral. Unless that integral admits a closed-form expression (or the latent variables are low-dimensional) it is intractable to compute.

To circumvent this intractability, investigators rely on approximate inference strategies such as variational Bayes (VB). VB approximates the posterior by solving an optimization problem. First propose an approximating family of distributions that contains all factorizable densities; then find the member of this family that minimizes the KL divergence to the (computationally intractable) exact posterior. Take this minimizer as a substitute for the posterior and carry out downstream data analysis.

VB scales to large datasets and works empirically in many difficult models. However, it comes with few theoretical guarantees, most of which focus on well-specified models. For example, Wang & Blei [31] establish the consistency and asymptotic normality of the VB posterior, assuming the data is generated by the probabilistic model. Under a similar assumption of a well-specified model, Zhang & Gao [36] derive the convergence rate of the VB posterior in settings with high-dimensional latent variables.

But as George Box famously quipped, "all models are wrong." Probabilistic models are rarely well-specified in practice. Does VB still enjoy good theoretical properties under model misspecification? What about the VB posterior predictive distributions? These are the questions we study in this paper.

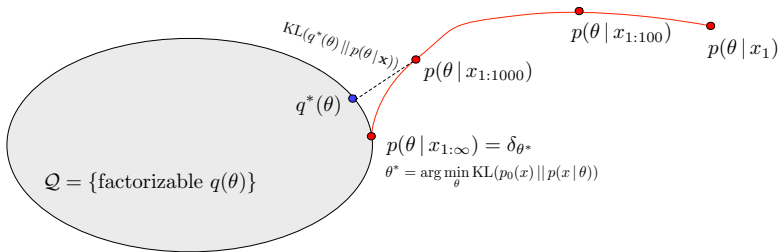

**Figure 1:** Why does the VB posterior converge to a point mass at $\theta^*$? The intuition behind this figure is described in § 1. In the figure, $q^*(x)$ is the optimal VB posterior given $x_{1:1000}$.

**Main idea.** We study VB under model misspecification. Under suitable conditions, we show that (1) the VB posterior is asymptotically normal, centering at the value that *minimizes* the KL divergence from the true distribution; (2) the VB posterior mean centers at the same value and is asymptotically normal; (3) in the variational posterior predictive, the error due to model misspecification dominates the error due to the variational approximation.

Concretely, consider $n$ data points $x_{1:n}$ independently and identically distributed with a true density $\prod_{i=1}^n p_0(x_i)$. Further consider a parametric probabilistic model with a $d$-dimensional latent variable $\theta = \theta_{1:d}$; its density belongs to the family $\{\prod_{i=1}^n p(x_i \mid \theta) : \theta \in \mathbb{R}^d\}$.[1] When the model is misspecified, it does not contain the true density, $p_0(x) \notin \{p(x \mid \theta) : \theta \in \Theta\}$.

Placing a prior $p(\theta)$ on the latent variable $\theta$, we infer its posterior $p(\theta \mid x_{1:n})$ using VB. Mean-field VB considers an approximating family $\mathcal{Q}$ that includes all factorizable densities

$$\mathcal{Q} = \left\{ q(\theta) : q(\theta) = \textstyle\prod_{i=1}^d q_i(\theta_i) \right\}.$$

It then finds the member that minimizes the KL divergence to the exact posterior $p(\theta \mid x_{1:n})$,

$$q^*(\theta) = \underset{q \in \mathcal{Q}}{\arg\min} \, \mathrm{KL}(q(\theta) || p(\theta \mid x_{1:n})). \tag{1}$$

The global minimizer $q^*(\theta)$ is called the VB posterior. (Here we focus on mean-field VB. The results below apply to VB with more general approximating families as well.)

We first study the asymptotic properties of the VB posterior and its mean. Denote $\theta^*$ as the value of $\theta$ that minimizes the KL divergence to the true distribution,

$$\theta^* = \underset{\theta}{\arg\min} \, \mathrm{KL}(p_0(x) || p(x \mid \theta)). \tag{2}$$

Note this KL divergence is different from the variational objective (Eq. 1); it is a property of the model class's relationship to the true density. We show that, under standard conditions, the VB posterior $q^*(\theta)$ converges in distribution to a point mass at $\theta^*$. Moreover, the VB posterior of the rescaled and centered latent variable $\tilde{\theta} = \sqrt{n}(\theta - \theta^*)$ is asymptotically normal. Similar asymptotics hold for the VB posterior mean $\hat{\theta}_{\mathrm{VB}} = \int \theta \cdot q^*(\theta) \, \mathrm{d}\theta$: it converges almost surely to $\theta^*$ and is asymptotically normal.

Why does the VB posterior converge to a point mass at $\theta^*$? The reason rests on three observations. (1) The classical Bernstein–von Mises theorem under model misspecification [18] says that the exact posterior $p(\theta \mid x_{1:n})$ converges to a point mass at $\theta^*$. (2) Because point masses are factorizable, this limiting exact posterior belongs to the approximating family $\mathcal{Q}$: if $\theta^* = (\theta_1^*, \theta_2^*, \theta_3^*)$, then $\delta_{\theta^*}(\theta) = \delta_{\theta_1^*}(\theta_1) \cdot \delta_{\theta_2^*}(\theta_2) \cdot \delta_{\theta_3^*}(\theta_3)$. (3) VB seeks the member in $\mathcal{Q}$ that is closest to the exact posterior (which also belongs to $\mathcal{Q}$, in the limit). Therefore, the VB posterior also converges to a point mass at $\theta^*$. Figure 1 illustrates this intuition—as we see more data, the posterior gets closer to the variational family. We make this argument rigorous in § 2.

The asymptotic characterization of the VB posterior leads to an interesting result about the VB approximation of the posterior predictive. Consider two posterior predictive distributions under the misspecified model. The VB predictive density is formed with the VB posterior,

$$p_{\mathrm{VB}}^{\mathrm{pred}}(x_{\mathrm{new}} \mid x_{1:n}) = \int p(x_{\mathrm{new}} \mid \theta) \cdot q^*(\theta) \, \mathrm{d}\theta. \tag{3}$$

The exact posterior predictive density is formed with the exact posterior,

$$p_{\text{exact}}^{\text{pred}}(x_{\text{new}} \,|\, x_{1:n}) = \int p(x_{\text{new}} \,|\, \theta) \cdot p(\theta \,|\, x_{1:n}) \, \mathrm{d}\theta. \tag{4}$$

Now define the *model misspecification error* to be the total variation (TV) distance between the exact posterior predictive and the true density $p_0(x)$. (When the model is well-specified, it converges to zero [27].) Further define the *variational approximation error* is the TV distance between the variational predictive and the exact predictive; it measures the price of the approximation when using the VB posterior to form the predictive. Below we prove that the model misspecification error dominates the variational approximation error—the variational approximation error vanishes as the number of data points increases. This result explains a widely observed phenomenon: VB achieves comparable predictive accuracy as MCMC even though VB uses an approximating family [4, 5, 7, 20].

The contributions of this work are to generalize the variational Bernstein–von Mises theorem [31] to misspecified models and to further study the VB posterior predictive distribution. § 2.1 and 2.2 details the results around VB in parametric probabilistic models. § 2.3 generalizes the results to probabilistic models where the dimensionality of latent variables can grow with the number of data points. § 2.4 illustrates the results in three forms of model misspecification, including underdispersion and misspecification of the latent dimensionality. § 3 corroborates the theoretical findings with simulation studies on generalized linear mixed model (GLMM) and latent Dirichlet allocation (LDA).

**Related work.** This work draws on two themes around VB and model misspecification.

The first theme is a body of work on the theoretical guarantees of VB. Many researchers have studied the properties of VB posteriors on particular Bayesian models, including linear models [23, 34], exponential family models [28, 29], generalized linear mixed models [14, 15, 22], nonparametric regression [10], mixture models [30, 32], stochastic block models [3, 35], latent Gaussian models [25], and latent Dirichlet allocation [13]. Most of these works assume well-specified models, with a few exceptions including [23, 25, 26].

In another line of work, Wang & Blei [31] establish the consistency and asymptotic normality of VB posteriors; Zhang & Gao [36] derive their convergence rate; and Pati et al. [24] provide risk bounds of VB point estimates. Further, Alquier & Ridgway [1], Alquier et al. [2], Yang et al. [33] study risk bounds for variational approximations of Gibbs posteriors and fractional posteriors, Chérief-Abdellatif et al. [9] study VB for model selection in mixtures, Jaiswal et al. [17] study $\alpha$-Rényi-approximate posteriors, and Fan et al. [11] and Ghorbani et al. [13] study generalizations of VB via TAP free energy. Again, most of these works focus on well-specified models. In contrast, we focus on VB in general misspecified Bayesian models and characterize the asymptotic properties of the VB posterior and the VB posterior predictive. Note, when the model is well-specified, our results recover the variational Bernstein–von Mises theorem of [31], but we further generalize their theory and extend it to analyzing the posterior predictive distribution.

The second theme is about characterizing posterior distributions under model misspecification. Allowing for model misspecification, Kleijn et al. [18] establishes consistency and asymptotic normality of the exact posterior in parametric Bayesian models; Kleijn et al. [19] studies exact posteriors in infinite-dimensional Bayesian models. We leverage these results around exact posteriors to characterize VB posteriors and VB posterior predictive distributions under model misspecification.

## 2 Variational Bayes (VB) under model misspecification

§ 2.1 and 2.2 examine the asymptotic properties of VB under model misspecification and for parametric models. § 2.3 extends these results to more general models, where the dimension of the latent variables grows with the data. § 2.4 illustrates the results with three types of model misspecification.

### 2.1 The VB posterior and the VB posterior mean

We first study the VB posterior $q^*(\theta)$ and its mean $\hat{\theta}_{\text{VB}}$. Assume iid data from a density $x_i \sim p_0$ and a parametric model $p(x \,|\, \theta)$, i.e., a model where the dimension of the latent variables does not grow with the data. We show that the optimal variational distribution $q^*(\theta)$ (Eq. 1) is asymptotically normal and centers at $\theta^*$ (Eq. 2), which minimizes the KL between the model $p_\theta$ and the true data generating distribution $p_0$. The VB posterior mean $\hat{\theta}_{\text{VB}}$ also converges to $\theta^*$ and is asymptotically normal.

Before stating these asymptotic results, we make a few assumptions about the prior $p(\theta)$ and the probabilistic model $\{p(x \mid \theta) : \theta \in \Theta\}$. These assumptions resemble the classical assumptions in the Bernstein–von Mises theorems [18, 27].

**Assumption 1** (Prior mass). *The prior density $p(\theta)$ is continuous and positive in a neighborhood of $\theta^*$. There exists a constant $M_p > 0$ such that $|(\log p(\theta))''| \leq M_p e^{|\theta|^2}$.*

Assumption 1 roughly requires that the prior has some mass around the optimal $\theta^*$. It is a necessary assumption: if $\theta^*$ does not lie in the prior support then the posterior cannot be centered there. Assumption 1 also requires a tail condition on $\log p(\theta)$: the second derivative of $\log p(\theta)$ can not grow faster than $\exp(|\theta|^2)$. This is a technical condition that many common priors satisfy.

**Assumption 2** (Consistent testability). *For every $\epsilon > 0$ there exists a sequence of tests $\phi_n$ such that*

$$\int \phi_n(x_{1:n}) \prod_{i=1}^{n} p_0(x_i)\, \mathrm{d}x_{1:n} \to 0, \tag{5}$$

$$\sup_{\{\theta:||\theta-\theta^*||\geq\epsilon\}} \int (1 - \phi_n(x_{1:n})) \cdot \left[ \prod_{i=1}^{n} \frac{p(x_i \mid \theta)}{p(x_i \mid \theta^*)} p_0(x_i) \right] \mathrm{d}x_{1:n} \to 0. \tag{6}$$

Assumption 2 roughly requires $\theta^*$ to be the unique optimum of the KL divergence to the truth (Eq. 2). In other words, $\theta^*$ is identifiable from fitting the probabilistic model $p(x \mid \theta)$ to the data drawn from $p_0(x)$. To satisfy this condition, it suffices to have the likelihood ratio $p(x \mid \theta_1)/p(x \mid \theta_2)$ be a continuous function of $x$ for all $\theta_1, \theta_2 \in \Theta$ (Theorem 3.2 of [18]).

Assumption 1 and Assumption 2 are classical conditions required for the asymptotic normality of the exact posterior Kleijn et al. [18]. They ensure that, for every sequence $M_n \to \infty$,

$$\int_\Theta \mathbb{1}(||\theta - \theta^*|| > \delta_n M_n) \cdot p(\theta \mid x_{1:n})\, \mathrm{d}\theta \xrightarrow{P_0} 0, \tag{7}$$

for some constant sequence $\delta_n \to 0$. In other words, the exact posterior $p(\theta \mid x)$ occupies vanishing mass outside of the $\delta_n M_n$-sized neighborhood of $\theta^*$. We note that the sequence $\delta_n$ also plays a role in the following local asymptotic normality (LAN) assumption.

**Assumption 3** (Local asymptotic normality (LAN)). *For every compact set $K \subset \mathbb{R}^d$, there exist random vectors $\Delta_{n,\theta^*}$ bounded in probability and nonsingular matrices $V_{\theta^*}$ such that*

$$\sup_{h \in K} \left| \log \frac{p(x \mid \theta^* + \delta_n h)}{p(x \mid \theta^*)} - h^\top V_{\theta^*} \Delta_{n,\theta^*} + \frac{1}{2} h^\top V_{\theta^*} h \right| \xrightarrow{P_0} 0, \tag{8}$$

*where $\delta_n$ is a $d \times d$ diagonal matrix that describes how fast each dimension of the $\theta$ posterior converges to a point mass. We note that $\delta_n \to 0$ as $n \to \infty$.*

This is a key assumption that characterizes the limiting normal distribution of the VB posterior. The quantities $\Delta_{n,\theta^*}$ and $V_{\theta^*}$ determine the normal distribution that the VB posterior will converge to. The constant $\delta_n$ determines the convergence rate of the VB posterior to a point mass. Many parametric models with a differentiable likelihood satisfy LAN. We provide a more technical description on how to verify Assumption 3 in Appendix A.

With these assumptions, we establish the asymptotic properties of the VB posterior and the VB posterior mean.

**Theorem 1.** (Variational Bernstein–von Mises Theorem under model misspecification, parametric model version) *Under Assumptions 1 to 3,*

1. *The VB posterior converges to a point mass at $\theta^*$:*

$$q^*(\theta) \xrightarrow{d} \delta_{\theta^*}. \tag{9}$$

2. *Denote $\tilde{\theta} = \delta_n^{-1}(\theta - \theta^*)$ as the re-centered and re-scaled version of $\theta$. The VB posterior of $\tilde{\theta}$ is asymptotically normal:*

$$\left\| q^*(\tilde{\theta}) - \mathcal{N}(\tilde{\theta}\,;\, \Delta_{n,\theta^*}, V_{\theta^*}'^{-1})) \right\|_{\mathrm{TV}} \xrightarrow{P_0} 0. \tag{10}$$

*where $V_{\theta^*}'$ is diagonal and has the same diagonal terms as the exact posterior precision matrix $V_{\theta^*}$.*

3. *The VB posterior mean converges to $\theta^*$ almost surely:*

$$\hat{\theta}_{\mathrm{VB}} \overset{a.s.}{\to} \theta^*. \tag{11}$$

4. *The VB posterior mean is also asymptotically normal:*

$$\delta_n^{-1}(\hat{\theta}_{\mathrm{VB}} - \theta^*) \overset{d}{\to} \Delta_{\infty,\theta^*}, \tag{12}$$

*where $\Delta_{\infty,\theta^*}$ is the limiting distribution of the random vectors $\Delta_{n,\theta^*}$: $\Delta_{n,\theta^*} \overset{d}{\to} \Delta_{\infty,\theta^*}$. Its distribution is $\Delta_{\infty,\theta^*} \sim \mathcal{N}\left(0, V_{\theta^*}^{-1}\mathbb{E}_{P_0}\left[(\log p(x\,|\,\theta^*))'(\log p(x\,|\,\theta^*))'^{\top}\right]V_{\theta^*}^{-1}\right)$.*

*Proof sketch.* The proof structure of Theorem 1 mimics Wang & Blei [31] but extends it to allow for model misspecification. In particular, we take care of the extra technicality due to the difference between the true data-generating measure $p_0(x)$ and the probabilistic model we fit $\{p(x\,|\,\theta) : \theta \in \Theta\}$.

The proof proceeds in three steps:

1. Characterize the asymptotic properties of the exact posterior:

$$p(\theta\,|\,x) \overset{d}{\to} \delta_{\theta^*},$$
$$\left\|p(\tilde{\theta}\,|\,x) - \mathcal{N}(\Delta_{n,\theta^*}, V_{\theta^*}^{-1})\right\|_{\mathrm{TV}} \overset{P_0}{\to} 0.$$

This convergence is due to Assumptions 1 and 2, and the classical Bernstein–von Mises theorem under model misspecification [18].

2. Characterize the KL minimizer of the limiting exact posterior in the variational approximating family $\mathcal{Q}$:

$$\arg\min_{q\in\mathcal{Q}} \mathrm{KL}(q(\theta)\,||\,p(\theta\,|\,x)) \overset{d}{\to} \delta_{\theta^*},$$
$$\left\|\arg\min_{q\in\mathcal{Q}} \mathrm{KL}(q(\tilde{\theta})\,||\,p(\tilde{\theta}\,|\,x)) - \mathcal{N}(\tilde{\theta}\,;\,\Delta_{n,\theta^*}, V_{\theta^*}'^{-1})\right\|_{\mathrm{TV}} \overset{P_0}{\to} 0,$$

where $V'$ is diagonal and shares the same diagonal terms as $V$. The intuition of this step is due to the observation that the point mass is factorizable: $\delta_{\theta^*} \in \mathcal{Q}$. We prove it via bounding the mass outside a neighborhood of $\theta^*$ under the KL minimizer $q^*(\theta)$.

3. Show that the VB posterior approaches the KL minimizer of the limiting exact posterior as the number of data points increases:

$$\left\|q^*(\theta) - \arg\min_{q\in\mathcal{Q}^d} \mathrm{KL}(q(\cdot)||\delta_{\theta^*})\right\|_{\mathrm{TV}} \overset{P_0}{\to} 0.$$
$$\left\|q^*(\tilde{\theta}) - \arg\min_{q\in\mathcal{Q}^d} \mathrm{KL}(q(\cdot)||\mathcal{N}(\cdot\,;\,\Delta_{n,\theta^*}, V_{\theta^*}^{-1}))\right\|_{\mathrm{TV}} \overset{P_0}{\to} 0.$$

The intuition of this step is that if two distributions are close, then their KL minimizer should also be close. In addition, the VB posterior is precisely the KL minimizer to the exact posterior: $q^*(\theta) = \arg\min_{q\in\mathcal{Q}^d} \mathrm{KL}(q(\theta)||p(\theta\,|\,x))$. We leverage $\Gamma$-convergence to prove this claim.

These three steps establish the asymptotic properties of the VB posterior under model misspecification (Theorem 1.1 and Theorem 1.2): the VB posterior converges to $\delta_{\theta^*}$ and is asymptotically normal.

To establish the asymptotic properties of the VB posterior mean (Theorem 1.3 and Theorem 1.4), we follow the classical argument in Theorem 2.3 of Kleijn et al. [18], which leverages that the posterior mean is the Bayes estimator under squared loss. The full proof is in Appendix D. □

Theorem 1 establishes the asymptotic properties of the VB posterior under model misspecification: it is asymptotically normal and converges to a point mass at $\theta^*$, which minimizes the KL divergence

to the true data-generating distribution. It also shows that the VB posterior mean shares similar convergence and asymptotic normality.

Theorem 1 states that, in the infinite data limit, the VB posterior and the exact posterior converge to the same point mass. The reason for this coincidence is (1) the limiting exact posterior is a point mass and (2) point masses are factorizable and hence belong to the variational approximating family $\mathcal{Q}$. In other words, the variational approximation has a negligible effect on the limiting posterior.

Theorem 1 also shows that the VB posterior has a different covariance matrix from the exact posterior. The VB posterior has a diagonal covariance matrix but the covariance of the exact posterior is not necessarily diagonal. However, the inverse of the two covariance matrices match in their diagonal terms. This fact implies that the entropy of the limiting VB posterior is always smaller than or equal to that of the limiting exact posterior (Lemma 8 of Wang & Blei [31]), which echoes the fact that the VB posterior is under-dispersed relative to the exact posterior.

We remark that the under-dispersion of the VB posterior does not necessarily imply under-coverage of the VB credible intervals. The reason is that, under model misspecification, even the credible intervals of the exact posterior cannot guarantee coverage [18]. Depending on how the model is misspecified, the credible intervals derived from the exact posterior can be arbitrarily under-covering or over-covering. Put differently, under model misspecification, neither the VB posterior nor the exact posterior are reliable for uncertainty quantification.

Consider a well-specified model, where $p_0(x) = p(x \mid \theta_0)$ for some $\theta_0 \in \Theta$ and $\theta^* = \theta_0$. In this case, Theorem 1 recovers the variational Bernstein–von Mises theorem [31]. That said, Assumptions 2 and 3 are stronger than their counterparts for well-specified models; the reason is that $P_0$ is usually less well-behaved than $P_{\theta_0}$. Assumptions 2 and 3 more closely align with those required in characterizing the exact posteriors under misspecification (Theorem 2.1 of [18]).

## 2.2 The VB posterior predictive distribution

We now study the posterior predictive induced by the VB posterior. As a consequence of Theorem 1, the error due to model misspecification dominates the error due to the variational approximation.

Recall that $p_{\text{VB}}^{\text{pred}}(x_{\text{new}} \mid x_{1:n})$ is the VB posterior predictive (Eq. 3), $p_{\text{true}}^{\text{pred}}(x_{\text{new}} \mid x_{1:n})$ is the exact posterior predictive (Eq. 4), $p_0(\cdot)$ is the true data generating density, and the TV distance between two densities $q_1$ and $q_2$ is $\|q_1(x) - q_2(x)\|_{\text{TV}} \triangleq \frac{1}{2} \int |q_1(x) - q_2(x)| \, \mathrm{d}x$.

**Theorem 2.** *(The* VB *posterior predictive distribution) If the probabilistic model is misspecified, i.e.* $\|p_0(x) - p(x \mid \theta^*)\|_{\text{TV}} > 0$, *then the model approximation error dominates the variational approximation error:*

$$\frac{\left\| p_{\text{VB}}^{\text{pred}}(x_{\text{new}} \mid x_{1:n}) - p_{\text{exact}}^{\text{pred}}(x_{\text{new}} \mid x_{1:n}) \right\|_{\text{TV}}}{\left\| p_0(x_{\text{new}}) - p_{\text{exact}}^{\text{pred}}(x_{\text{new}} \mid x_{1:n}) \right\|_{\text{TV}}} \xrightarrow{P_0} 0, \tag{13}$$

*under the regularity condition* $\int \nabla_\theta^2 p(x \mid \theta^*) \, \mathrm{d}x < \infty$ *and Assumptions 1 to 3.*

*Proof sketch.* Theorem 2 is due to two observations: (1) in the infinite data limit, the VB posterior predictive converges to the exact posterior predictive and (2) in the infinite data limit, the exact posterior predictive does *not* converge to the true data-generating distribution because of model misspecification. Taken together, these two observations give Eq. 13.

The first observation comes from Theorem 1, which implies that both the VB posterior and the exact posterior converge to the same point mass $\delta_{\theta^*}$ in the infinite data limit. Thus, they lead to similar posterior predictive distributions, which gives

$$\left\| p_{\text{VB}}^{\text{pred}}(x_{\text{new}} \mid x_{1:n}) - p_{\text{true}}^{\text{pred}}(x_{\text{new}} \mid x_{1:n}) \right\|_{\text{TV}} \xrightarrow{P_0} 0. \tag{14}$$

Moreover, the model is assumed to be misspecified $\|p_0(x) - p(x \mid \theta^*)\|_{\text{TV}} > 0$, which implies

$$\left\| p_0(x_{\text{new}}) - p_{\text{exact}}^{\text{pred}}(x_{\text{new}} \mid x_{1:n}) \right\|_{\text{TV}} \rightarrow c_0 > 0. \tag{15}$$

This fact shows that the model misspecification error does not vanish in the infinite data limit. Eq. 14 and Eq. 15 imply Theorem 2. The full proof of Theorem 2 is in Appendix E. □

As the number of data points increases, Theorem 2 shows that the model misspecification error dominates the variational approximation error. The reason is that both the VB posterior and the exact posterior converge to the same point mass. So, even though the VB posterior has an under-dispersed covariance matrix relative to the exact posterior, both covariance matrices shrink to zero in the infinite data limit; they converge to the same posterior predictive distributions.

Theorem 2 implies that when the model is misspecified, VB pays a negligible price in its posterior predictive distribution. In other words, if the goal is prediction, we should focus on finding the correct model rather than on correcting the variational approximation. For the predictive ability of the posterior, the problem of an incorrect model outweighs the problem of an inexact inference.

Theorem 2 also explains the phenomenon that VB predicts well despite being an approximate inference method. As models are rarely correct in practice, the error due to model misspecification often dominates the variational approximation error. Thus, on large datasets, VB can achieve comparable predictive performance, even when compared to more exact Bayesian inference algorithms (like long-run MCMC) that do not use approximating families [4, 5, 7, 20].

## 2.3 Variational Bayes (VB) in misspecified general probabilistic models

§ 2.1 and 2.2 characterize the VB posterior, the VB posterior mean, and the VB posterior predictive distribution in misspecified parametric models. Here we extend these results to a more general class of (misspecified) models with both global latent variables $\theta = \theta_{1:d}$ and local latent variables $z = z_{1:n}$. This more general class allows the local latent variables to grow with the size of the data. The key idea is to reduce this class to the simpler parametric models, via what we call the "variational model."

Consider the following probabilistic model with both global and local latent variables for $n$ data points $x = x_{1:n}$,

$$p(\theta, x, z) = p(\theta) \prod_{i=1}^{n} p(z_i \mid \theta) p(x_i \mid z_i, \theta). \tag{16}$$

The goal is to infer $p(\theta \mid x)$, the posterior of the global latent variables.[2]

VB approximates the posterior of both global and local latent variables $p(\theta, z \mid x)$ by minimizing its KL to the exact posterior:

$$q^*(\theta)q^*(z) = q^*(\theta, z) = \underset{q \in \mathcal{Q}}{\arg\min} \, \mathrm{KL}(q(\theta, z) || p(\theta, z \mid x)), \tag{17}$$

where $\mathcal{Q} = \{q : q(\theta, z) = \prod_{i=1}^{d} q_{\theta_i}(\theta_i) \prod_{j=1}^{n} q_{z_j}(z_j)\}$ is the approximating family that contains all factorizable densities. (The first equality is because $q^*(\theta, z)$ belongs to the factorizable family $\mathcal{Q}$.) The VB posterior of the global latent variables $\theta_{1:d}$ is $q^*(\theta)$.

VB for general probabilistic models operates in the same way as for parametric models, except we must additionally approximate the posterior of the local latent variables. Our strategy is to reduce the general probabilistic model with VB to a parametric model (§ 2.1). Consider the so-called variational log-likelihood [31],

$$\log p^{\mathrm{VB}}(x \mid \theta) = \eta(\theta) + \max_{q(z) \in \mathcal{Q}} \mathbb{E}_{q(z)} \left[ \log p(x, z \mid \theta) - \log q(z) \right], \tag{18}$$

where $\eta(\theta)$ is a log normalizer. Now construct the *variational model* with $p^{\mathrm{VB}}(x \mid \theta)$ as the likelihood and $\theta$ as the global latent variable. This model no longer contains local latent variables; it is a parametric model.

Using the same prior $p(\theta)$, the variational model leads to a posterior on the global latent variable

$$\pi^*(\theta \mid x) \triangleq \frac{p(\theta) p^{\mathrm{VB}}(x \mid \theta)}{\int p(\theta) p^{\mathrm{VB}}(x \mid \theta) \, \mathrm{d}\theta}. \tag{19}$$

As shown in [31], the VB posterior, which optimizes the variational objective, is close to $\pi^*(\theta \mid x)$,

$$q^*(\theta) = \underset{q \in \mathcal{Q}}{\arg\min} \, \mathrm{KL}(q(\theta) || \pi^*(\theta \mid x)) + o_{P_0}(1). \tag{20}$$

Notice that Eq. 20 resembles Eq. 1. This observation leads to a reduction of VB in general probabilistic models to VB in parametric probabilistic models with an alternative likelihood $p^{\mathrm{VB}}(x \mid \theta)$. This perspective then allows us to extend Theorems 1 and 2 in § 2.1 to general probabilistic models.

More specifically, we define the optimal value $\theta^*$ as in parametric models:

$$\theta^* \triangleq \arg\max \mathrm{KL}(p_0(x) || p^{\mathrm{VB}}(\theta \, ; \, x)). \tag{21}$$

This definition of $\theta^*$ coincides with the definition in parametric models (Eq. 2) when the model is indeed parametric.

Next we state the assumptions and results for the VB posterior and the VB posterior mean for general probabilistic models.

**Assumption 4** (Consistent testability). *For every $\epsilon > 0$ there exists a sequence of tests $\phi_n$ such that*

$$\int \phi_n(x) p_0(x) \, \mathrm{d}x \to 0, \tag{22}$$

$$\sup_{\{\theta : ||\theta - \theta^*|| \geq \epsilon\}} \int (1 - \phi_n(x)) \frac{p^{\mathrm{VB}}(x \mid \theta)}{p^{\mathrm{VB}}(x \mid \theta^*)} p_0(x) \, \mathrm{d}x \to 0. \tag{23}$$

**Assumption 5** (Local asymptotic normality (LAN)). *For every compact set $K \subset \mathbb{R}^d$, there exist random vectors $\Delta_{n,\theta^*}$ bounded in probability and nonsingular matrices $V_{\theta^*}$ such that*

$$\sup_{h \in K} \left| \log \frac{p^{\mathrm{VB}}(x \mid \theta^* + \delta_n h)}{p^{\mathrm{VB}}(x \mid \theta^*)} - h^\top V_{\theta^*} \Delta_{n,\theta^*} + \frac{1}{2} h^\top V_{\theta^*} h \right| \xrightarrow{P_0} 0, \tag{24}$$

*where $\delta_n$ is a $d \times d$ diagonal matrix, where $\delta_n \to 0$ as $n \to \infty$.*

Assumptions 4 and 5 are analogous to Assumptions 2 and 3 except that we replace the model $p(x \mid \theta)$ with the variational model $p^{\mathrm{VB}}(x \mid \theta)$. In particular, Assumption 6 is a LAN assumption on probabilistic models with local latent variables, i.e. nonparametric models. While the LAN assumption does not hold generally in nonparametric models with infinite-dimensional parameters [12], there are a few nonparametric models that have been shown to satisfy the LAN assumption, including generalized linear mixed models [15], stochastic block models [3], and mixture models [32]. We illustrate how to verify Assumptions 4 and 5 for specific models in Appendix C. We refer the readers to Section 3.4 of Wang & Blei [31] for a detailed discussion on these assumptions about the variational model.

Under Assumptions 1, 4 and 5, Theorems 1 and 2 can be generalized to general probabilistic models. The full details of these results (Theorems 3 and 4) are in Appendix B.

## 2.4 Applying the theory

To illustrate the theorems, we apply Theorems 1, 2, 3 and 4 to three types of model misspecification: underdispersion in Bayesian regression of count data, component misspecification in Bayesian mixture models, and latent dimensionality misspecification with Bayesian stochastic block models. For each model, we verify the assumptions of the theorems and then characterize the limiting distribution of their VB posteriors. The details of these results are in Appendix C.

## 3 Simulations

We illustrate the implications of Theorems 1, 2, 3 and 4 with simulation studies. We studied two models, Bayesian GLMM [21] and LDA [6]. To make the models misspecified, we generate datasets from an "incorrect" model and then perform approximate posterior inference. We evaluate how close the approximate posterior is to the limiting exact posterior $\delta_{\theta^*}$, and how well the approximate posterior predictive captures the test sets.

To approximate the posterior, we compare VB with Hamiltonian Monte Carlo (HMC), which draws samples from the exact posterior. We find that both achieve similar closeness to $\delta_{\theta^*}$ and comparable predictive log likelihood on test sets. We use two automated inference algorithms in Stan [8]:

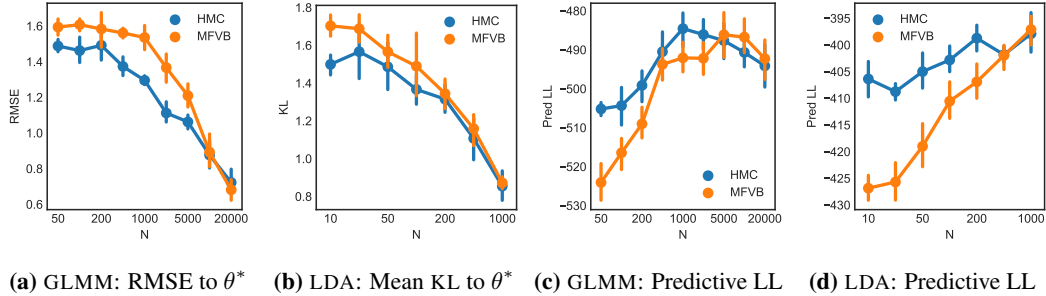

**(a)** GLMM: RMSE to $\theta^*$    **(b)** LDA: Mean KL to $\theta^*$    **(c)** GLMM: Predictive LL    **(d)** LDA: Predictive LL

**Figure 2:** Dataset size versus closeness to the limiting exact posterior $\delta_{\theta^*}$ and posterior predictive log likelihood on test data (mean $\pm$ sd). VB posteriors and MCMC posteriors achieve similar closeness to $\delta_{\theta^*}$ and comparable predictive accuracy.

automatic differentiation variational inference (ADVI) [20] for VB and No-U-Turn sampler (NUTS) [16] for HMC. We lay out the detailed simulation setup in Appendix I.

**Bayesian GLMM.** We simulate data from a negative binomial linear mixed model (LMM): each individual belongs to one of the ten groups; each group has $N$ individuals; and the outcome is affected by a random effect due to this group membership. Then we fit a Poisson LMM with the same group structure, which is misspecified with respect to the simulated data. Figure 2a shows that the RMSE to $\theta^*$ for the VB and MCMC posterior converges to similar values as the number of individuals increases. This simulation corroborates Theorems 1 and 3: the limiting VB posterior coincide with the limiting exact posterior. Figure 2c shows that VB and MCMC achieve similar posterior predictive log likelihood as the dataset size increases. It echoes Theorems 2 and 4: when performing prediction, the error due to the variational approximation vanishes with infinite data.

**Latent Dirichlet allocation (LDA).** We simulate $N$ documents from a 15-dimensional LDA and fit a 10-dimensional LDA; the latent dimensionality of LDA is misspecified. Figure 2b shows the distance between the VB/MCMC posterior topics to the limiting exact posterior topics, measured by KL averaged over topics. When the number of documents is at least 200, both VB and MCMC are similarly close to the limiting exact posterior. Figure 2d shows that, again once there are 200 documents, the VB and MCMC posteriors also achieve similar predictive ability. These results are consistent with Theorems 1, 2, 3 and 4.

## 4 Discussion

In this work, we study VB under model misspecification. We show that the VB posterior is asymptotically normal, centering at the value that *minimizes* the KL divergence from the true distribution. The VB posterior mean also centers at the same value and is asymptotically normal. These results generalize the variational Bernstein–von Mises theorem Wang & Blei [31] to misspecified models. We further study the VB posterior predictive distributions. We find that the model misspecification error dominates the variational approximation error in the VB posterior predictive distributions. These results explain the empirical phenomenon that VB predicts comparably well as MCMC even if it uses an approximating family. It also suggests that we should focus on finding the correct model rather than de-biasing the variational approximation if we use VB for prediction.

An interesting direction for future work is to characterize local optima of the evidence lower bound (ELBO), which is the VB posterior we obtain in practice. The results in this work all assume that the ELBO optimization returns global optima. It provides the possibility for local optima to share these properties, though further research is needed to understand the precise properties of local optima. Combining this work with optimization guarantees may lead to a fruitful further characterization of variational Bayes.

**Acknowledgments.** We thank Victor Veitch and Jackson Loper for helpful comments on this article. This work is supported by ONR N00014-17-1-2131, ONR N00014-15-1-2209, NIH 1U01MH115727-01, NSF CCF-1740833, DARPA SD2 FA8750-18-C-0130, IBM, 2Sigma, Amazon, NVIDIA, and Simons Foundation.

## Footnotes

[1] A parametric probabilistic model means the dimensionality of the latent variables do not grow with the number of data points. We extend these results to more general probabilistic models in § 2.3.

[2]This model has one local latent variable per data point. But the results here extend to probabilistic models with $z = z_{1:d_n}$ and non i.i.d data $x = x_{1:n}$. We only require that $d$ stays fixed as $n$ grows but $d_n$ grows with $n$.

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
