[Supplementary Material · vbmisspec_supp.pdf]

# Supplementary Material: Variational Bayes under Model Misspecification

## A  The LAN assumption (Assumption 3) in parametric models

The LAN of parametric models (i.e. when parametric models satisfy Assumption 3) has been widely studied in the literature [4, 6, 10]. For example, it suffices for a parametric model to satisfy: (1) the log density $\log p(x \mid \theta)$ is differentiable at $\theta^*$, (2) the log likelihood ratio is bounded by some square integrable function $m_{\theta^*}(x)$: $|\log \frac{p(x \mid \theta_1)}{p(x \mid \theta_2)}| \le m_{\theta^*}(x) \|\theta_1 - \theta_2\|$ $P_0$-almost surely, and (3) the KL divergence has a second order Taylor expansion around $\theta^*$: $-\int p_0(x) \log \frac{p(x \mid \theta)}{p(x \mid \theta^*)} = \frac{1}{2}(\theta - \theta^*)V_{\theta^*}(\theta - \theta^*) + o(\|\theta - \theta^*\|)$ when $\theta \to \theta^*$ (Lemma 2.1 of Kleijn et al. [6]). Under these conditions, the parametric model satisfies Assumption 3 with $\delta_n = (\sqrt{n})^{-1}$, which leads to the $\sqrt{n}$-convergence of the exact posterior.

## B  Extending Theorems 1 and 2 to general probabilistic models

We study VB in general probabilistic models via the above reduction with the variational model $p^{\mathrm{VB}}(x \mid \theta)$. We posit analogous assumptions on $p^{\mathrm{VB}}(x \mid \theta)$ as Assumptions 2 and 3 and extend the results in § 2.1 to general probabilistic models. Consider $n$ data points $x = x_{1:n}$, but they no longer need to be i.i.d. We define the true data-generating density of $x$ as $p_0(x)$.

We state the asymptotic properties of the VB posterior in general probabilistic models.

**Theorem 3.** (Variational Bernstein–von Mises Theorem under model misspecification) *Under Assumptions 1, 4 and 5,*

1. *The VB posterior converges to a point mass at $\theta^*$:*

$$q^*(\theta) \xrightarrow{d} \delta_{\theta^*}. \tag{25}$$

2. *Denote $\tilde{\theta} = \delta_n^{-1}(\theta - \theta^*)$ as the re-centered and re-scaled version of $\theta$. The VB posterior of $\tilde{\theta}$ is asymptotically normal:*

$$\left\| q^*(\tilde{\theta}) - \mathcal{N}(\tilde{\theta}\,;\, \Delta_{n,\theta^*}, V_{\theta^*}'^{-1})) \right\|_{\mathrm{TV}} \xrightarrow{P_0} 0. \tag{26}$$

   *where $V_{\theta^*}'$ is diagonal and has the same diagonal terms as $V_{\theta^*}$.*

3. *Denote $\hat{\theta}_{\mathrm{VB}} = \int \theta \cdot q^*(\theta)\,\mathrm{d}\theta$ as the mean of the VB posterior. The VB posterior mean converges to $\theta^*$ almost surely:*

$$\hat{\theta}_{\mathrm{VB}} \xrightarrow{a.s.} \theta^*. \tag{27}$$

4. *The VB posterior mean is also asymptotically normal:*

$$\delta_n^{-1}(\hat{\theta}_{\mathrm{VB}} - \theta^*) \xrightarrow{d} \Delta_{\infty,\theta^*}, \tag{28}$$

   *where $\Delta_{\infty,\theta^*}$ is the limiting distribution of the random vectors $\Delta_{n,\theta^*}$: $\Delta_{n,\theta^*} \xrightarrow{d} \Delta_{\infty,\theta^*}$. Its distribution is $\Delta_{\infty,\theta^*} \sim \mathcal{N}\left(0, V_{\theta^*}^{-1}\mathbb{E}_{P_0}\left[(\log p^{\mathrm{VB}}(x \mid \theta^*))'(\log p^{\mathrm{VB}}(x \mid \theta^*))'^\top\right] V_{\theta^*}^{-1}\right)$.*

Theorem 3 repeats Theorem 1 except that the limiting distribution of the VB posterior mean is governed by $p^{\mathrm{VB}}(x \mid \theta^*)$ as opposed to $p(x \mid \theta^*)$. Theorem 3 reduces to Theorem 1 when the probabilistic model we fit is parametric.

With an additional LAN assumption on the probabilistic model, we can further extend the characterization of the VB posterior predictive distribution (Theorem 2) to general probabilistic models.

**Assumption 6** (LAN). *For every compact set $K \subset \mathbb{R}^d$, there exist random vectors $\Delta_{n,\theta^*}^0$ bounded in probability and nonsingular matrices $V_{\theta^*}^0$ such that*

$$\sup_{h \in K} \left| \log \frac{p(x \mid \theta^* + \delta_n h)}{p(x \mid \theta^*)} - h^\top V_{\theta^*}^0 \Delta_{n,\theta^*}^0 + \frac{1}{2} h^\top V_{\theta^*}^0 h \right| \xrightarrow{P_0} 0. \tag{29}$$

Assumption 6 requires that the probability model $p(x \mid \theta) = \int p(x, z \mid \theta) \, dz$ has a LAN expansion at $\theta^*$. Many models satisfy Assumption 6, including Gaussian mixture models [13], Poisson linear mixed models [5], stochastic block models [1]. We note that the LAN expansion of the model $p(x \mid \theta)$ can be different from that of the variational model $p^{\mathrm{VB}}(x \mid \theta^*)$.

**Theorem 4.** *(The* VB *posterior predictive distribution) If the probabilistic model is misspecified, i.e.* $\|p_0(x) - p(x \mid \theta^*)\|_{\mathrm{TV}} > 0$, *then the model approximation error dominates the variational approximation error:*

$$\frac{\left\| p_{\mathrm{VB}}^{\mathrm{pred}}(x_{\mathrm{new}} \mid x) - p_{\mathrm{exact}}^{\mathrm{pred}}(x_{\mathrm{new}} \mid x) \right\|_{\mathrm{TV}}}{\left\| p_0(x_{\mathrm{new}}) - p_{\mathrm{exact}}^{\mathrm{pred}}(x_{\mathrm{new}} \mid x) \right\|_{\mathrm{TV}}} \xrightarrow{P_0} 0, \tag{30}$$

*assuming* $\int \nabla_\theta^2 p(x \mid \theta^*) \, dx < \infty$ *and Assumptions 1, 4, 5 and 6. Notation wise,* $p_{\mathrm{VB}}^{\mathrm{pred}}(x_{\mathrm{new}}) = \int p(x_{\mathrm{new}} \mid \theta) q^*(\theta) \, d\theta$ *is the* VB *posterior predictive density,* $p_{\mathrm{true}}^{\mathrm{pred}}(x_{\mathrm{new}}) = \int p(x_{\mathrm{new}} \mid \theta) p(\theta \mid x) \, d\theta$ *is the exact posterior predictive density,* $p(x \mid \theta) = \int p(x, z \mid \theta) \, dz$ *is the marginal density of the model, and* $p_0(\cdot)$ *is the true data generating density.*

Theorem 3 and Theorem 4 generalizes the asymptotic characterizations of the VB posterior, the VB posterior mean, and the VB posterior predictive distributions to general probabilistic models. As in parametric probabilistic models, the VB posterior and its mean both remain asymptotically normal and centered at $\theta^*$ in general probabilistic models. The model misspecification error continues to dominate the variational approximation error in the VB posterior predictive distributions.

The proofs of Theorem 3 and Theorem 4 extends those of Theorem 1 and Theorem 2 by leveraging the connection between the variational model and the original model (Eq. 20). The full proofs are in Appendix D and Appendix E.

## C  Applications of Theorems 1, 2, 3 and 4

We illustrate Theorems 1, 2, 3 and 4 with three types of model misspecification: under-dispersion in generalized linear model, component misspecification in Bayesian mixture models, and latent dimensionality misspecification in Bayesian stochastic block models. In all three cases, we show that the VB posterior converges to a point mass at the value that minimizes the KL divergence to the true data generating distribution; their VB posterior predictive distributions also converge to the true posterior predictive distributions.

### C.1  Under-dispersion in Bayesian count regression

The first type of model misspecification we consider is under-dispersion. Suppose the data is generated by a Negative Binomial regression model but we fit Poisson regression. We characterize the asymptotic properties of the VB posteriors of the coefficient.

**Corollary 5.** *Consider data* $\mathcal{D} = \{(X_i, Y_i)\}_{i=1}^n$. *Assume the data generating measure* $P_0$ *has density*

$$p_0(Y_i \mid X_i) = \mathrm{NegativeBinomial}\left( r, \frac{\exp(X_i^\top \beta_0)}{1 + \exp(X_i^\top \beta_0)} \right)$$

*for some constant* $r$. *Let* $q^*(\beta)$ *and* $\hat{\beta}$ *denote the* VB *posterior and its mean. We fit a Poisson regression model to the data* $\mathcal{D}$

$$p(Y_i \mid X_i, \beta) = \mathrm{Poisson}(\exp(X_i^\top \beta)),$$

*with a prior that satisfies Assumption 1. Let* $\beta^*$ *be the value such that*

$$\sum_{i=1}^n \mathbb{E}_{P_0}\left[ Y_i - \exp(X_i^\top \beta^*) X_i \mid X_i \right] = 0.$$

*Then we have*

$$\left\| q^*(\beta) - \mathcal{N}\left( \beta \, ; \, \beta^*, \frac{1}{n} V'^{-1} \right) \right\|_{\mathrm{TV}} \xrightarrow{P_0} 0,$$

*and*

$$\sqrt{n}(\hat{\beta} - \beta^*) \xrightarrow{d} \mathcal{N}(0, V'^{-1}),$$

*where* $V' = \text{diag} \left[ (X \exp(\frac{1}{2} X^\top \beta^*))^\top (X \exp(\frac{1}{2} X^\top \beta^*)) \right]$. *Moreover, the model misspecification error dominates the* VB *approximation error in the* VB *posterior predictive distribution.*

*Proof.* First, we notice that

$$\beta^* = \arg\min_{\beta} \text{KL}(p_0(Y \mid X) \,\|\, p(Y \mid X, \beta)).$$

We then verify Assumptions 2 and 3. Assumption 2 is satisfied because the maximum likelihood estimator converges to $\beta^*$ [8]. Assumption 3 is satisfied because the log likelihood of Poisson likelihood is differentiable. Moreover, the log likelihood ratio is bounded a squared integrable function and have a second order Taylor expansion under $P_0$. Hence Lemma 2.1 of Kleijn et al. [6] implies Assumption 3. Given Assumptions 1 to 3, we apply Theorems 1 and 2 and conclude Corollary 5. □

### C.2 Component misspecification in Bayesian mixture models

The second type of misspecification we consider is component misspecification. Suppose the data is generated by a Bayesian mixture model where each component is Gaussian and shares the same variance. But we fit a Bayesian mixture model where each component is a $t$-distribution. We characterize the asymptotic properties of the VB posteriors.

**Corollary 6.** *Consider data* $\mathcal{D} = \{(X_i)\}_{i=1}^n$. *Assume the data generating measure* $P_0$ *has density*

$$p_0(x_i) = \sum_{c_i \in \{1,\ldots,K\}} \mathcal{N}(x_i \mid \mu_{c_i}, \Sigma) \cdot \text{Categorical}(c_i \,;\, 1/K),$$

*where* $\Sigma$ *and* $K$ *are two constants. Consider a mixture model where each component is* $t$-*distribution with* $\nu$ *degree of freedom,*

$$p(x_i \mid \theta) = \sum_{c_i \in \{1,\ldots,K\}} t(x_i \mid \theta_{c_i}, \nu) \cdot \text{Categorical}(c_i \,;\, 1/K),$$

*with priors that satisfy Assumption 1. The goal is to estimate the component centers* $\theta \triangleq (\theta_1, \ldots, \theta_K)$. *Let* $q^*(\theta)$ *and* $\hat{\theta}$ *denote the* VB *posterior and its mean. Further denote the variational log likelihood as follows:*

$$m(\theta \,;\, x) = \sup_{q(c) \in \mathcal{Q}^n} \int q(c) \log \frac{p(x, c \mid \theta)}{q(c)} \, dc.$$

*Under regularity conditions (A1-A5) and (B1,2,4) of Westling & McCormick [13], we have*

$$\left\| q^*(\theta) - \mathcal{N}\left(\theta^* + \frac{Y}{\sqrt{n}}, \frac{1}{n} V_0(\theta^*)\right) \right\|_{\text{TV}} \xrightarrow{P_0} 0,$$

*and*

$$\sqrt{n}(\hat{\theta} - \theta^*) \xrightarrow{d} Y,$$

*where* $\theta^*$ *satisfies*

$$\mathbb{E}_{P_0} \nabla_\theta^2 m(\theta^* \,;\, x) = 0.$$

*The limiting distribution* $Y$ *is* $Y \sim \mathcal{N}(0, V(\theta^*))$, *where* $V(\theta^*) = A(\theta^*)^{-1} B(\theta^*) A(\theta^*)^{-1}$, $A(\theta) = \mathbb{E}_{P_0}[\nabla_\theta^2 m(\theta \,;\, x)]$, *and* $B(\theta) = \mathbb{E}_{P_0}[\nabla_\theta m(\theta \,;\, x) \nabla_\theta m(\theta \,;\, x)^\top]$. *The diagonal matrix* $V_0(\theta^*)$ *satisfies* $(V_0(\theta^*)^{-1})_{ii} = (A(\theta^*))_{ii}$. *Moreover, the model misspecification error dominates the* VB *approximation error in the* VB *posterior predictive distribution.*

*The specification of a mixture model is invariant to permutation among* $K$ *components; this corollary is true up to permutations among the* $K$ *components.*

*Proof.* First, we notice that $\theta^* = \arg\min_\theta \text{KL}(p_0(X) \,\|\, p(X \mid \theta))$. We then verify Assumptions 4, 5 and 6. Assumption 4 is satisfied because the variational log likelihood $m(x \,;\, \theta)$ yields consistent estimates of $\theta$ [13]. Assumptions 5 and 6 is satisfied by a standard Taylor expansion of $m(x \,;\, \theta)$ and $p(x \mid \theta)$ at $\theta^*$ (Eq. 128 of Wang & Blei [11]). Given Assumptions 1, 4 and 5, we apply Theorems 3 and 4 and conclude Corollary 6. □

## C.3 Latent dimensionality misspecification in Bayesian Stochastic Block Models

The third type of misspecification we consider is latent dimensionality misspecification. Suppose the data is generated by a Bayesian stochastic block model with $K$ communities. But we fit a Bayesian stochastic block model with only $K-1$ communities. We characterize the asymptotic properties of the VB posteriors.

**Corollary 7.** *Consider the adjacency matrix $\mathcal{D} = \{(X_{ij})\}_{i,j=1}^n$ of a network with $n$ nodes. Suppose it is generated from the stochastic block model with $K$ communities and parameters $\nu$ and $\omega$. The parameter $\nu$ represents the odds ratio for a node to belong to each of the $K$ communities. For simplicity, we assume the odds ratio is the same for all the $K$ communities. The parameter $\omega$ represents the $K \times K$ matrix of odds ratios; the $(i,j)$-entry of $\omega$ is the odds ratio of two nodes being connected if they belong to community $i$ and $j$ respectively.*

*We fit a stochastic block model with $K-1$ communities, whose prior satisfies Assumption 1. Let $q_\nu^*(\nu_{(k-1)}), q_\omega^*(\omega_{(k-1)})$ denote the VB posterior of $\nu_{(k-1)}$ and $\omega_{(k-1)}$, where $\nu_{(k-1)}$ and $\omega_{(k-1)}$ are the odds ratios vector and matrix for the $(K-1)$-dimensional stochastic block model. Similarly, let $\hat{\nu}, \hat{\omega}$ be the VB posterior mean. Then*

$$\left\| q_\nu^*(\nu)q_\omega^*(\omega) - \mathcal{N}\left((\nu,\omega);(\nu^*,\omega^*) + (\frac{\Sigma_1^{-1}Y_1}{\sqrt{n\lambda_0}}, \frac{\Sigma_2^{-1}Y_2}{\sqrt{n}}), V_n(\nu^*,\omega^*)\right) \right\|_{\mathrm{TV}} \xrightarrow{P_0} 0$$

*where $\omega^*(a) = \log \frac{\pi_{(k-1)}(a)}{1-\sum_{b=1}^{K-2}\pi_{(k-1)}(b)}$, $\nu^*(a,b) = \log \frac{H_{(k-1)}(a,b)}{1-H_{(k-1)}(a,b)}, a, b = 1, \ldots, K-1$, where $\pi_{(k-1)}(a)$ and $H_{(k-1)}(a,b)$ are the community weights vector and the connectivity matrix where two smallest communities are merged. The constant $\lambda_0$ is $\lambda_0 = \mathbb{E}_{P_0}(\text{degree of each node})$, $(\log n)^{-1}\lambda_0 \to \infty$. $Y_1$ and $Y_2$ are two zero mean random vectors with covariance matrices $\Sigma_1$ and $\Sigma_2$, where $\Sigma_1, \Sigma_2$ are known functions of $\nu^*, \omega^*$. The diagonal matrix $V(\nu^*, \omega^*)$ satisfies $V^{-1}(\nu^*, \omega^*)_{ii} = diag(\Sigma_1, \Sigma_2)_{ii}$. Also,*

$$(\sqrt{n\lambda_0}(\hat{\nu} - \nu^*), \sqrt{n}(\hat{\omega} - \omega^*)) \xrightarrow{d} (\Sigma_1^{-1}Y_1, \Sigma_2^{-1}Y_2),$$

*Moreover, the model misspecification error dominates the VB approximation error in the VB posterior predictive distribution.*

*The specification of classes in stochastic block model (SBM) is permutation invariant. So the convergence above is true up to permutation with the $K-1$ classes. We follow Bickel et al. [1] to consider the quotient space of $(\nu, \omega)$ over permutations.*

*Proof.* First, we notice that $(\nu^*, \omega^*) = \arg\min_{\nu,\omega} \mathrm{KL}(p_0(X_{ij}) \,||\, p(X_{ij} \,|\, \nu, \omega))$. We then verify Assumptions 4, 5 and 6. Assumption 4 is satisfied because the variational log likelihood of stochastic block models yields consistent estimates of $\nu, \omega$ even under under-fitted model [1, 12]. Assumptions 5 and 6 is established by Lemmas 2,3, and Theorem 3 of [1]. Given Assumptions 1, 4 and 5, we apply Theorems 3 and 4 and conclude Corollary 7. $\square$

# D  Proof of Theorems 1 and 3

We prove Theorem 3 in this section. Theorem 1 follows directly from Theorem 3 in parametric models.

*Proof.* The proof of Theorem 3 mimics the proof structure of Wang & Blei [11] except we need to take care of the additional technical complications due to model misspecification.

We first study the VB ideal $\pi^*(\theta \mid x)$, defined as

$$\pi^*(\theta \mid x) \triangleq \frac{p(\theta)p^{\mathrm{VB}}(x \,;\, \theta)}{\int p(\theta)p^{\mathrm{VB}}(x \,;\, \theta)\,\mathrm{d}\theta}.$$

The VB ideal $\pi^*(\theta \mid x)$ is the posterior of $\theta$ if we only perform variational approximation on the local latent variables. In other words, it is the exact posterior of the variational model $p^{\mathrm{VB}}(x \,;\, \theta)$.

We note that the VB ideal is different from the VB posterior. However, we will show later that the two are closely related.

Consider a change-of-variable (re-centering and re-scaling) of $\theta$ into $\tilde{\theta} = \delta_n^{-1}(\theta - \theta^*)$. Lemma 8 shows that the posterior of $\tilde{\theta}$ is close to $\mathcal{N}(\cdot; \Delta_{n,\theta^*}, V_{\theta^*}^{-1})$, where $\Delta_{n,\theta^*}$ and $V_{\theta^*}^{-1}$ are two constants in Assumption 5.

**Lemma 8.** *The* VB *ideal converges in total variation to a sequence of normal distributions,*

$$||\pi_{\tilde{\theta}}^*(\cdot \mid x) - \mathcal{N}(\cdot; \Delta_{n,\theta^*}, V_{\theta^*}^{-1})||_{\mathrm{TV}} \xrightarrow{P_0} 0.$$

The proof of Lemma 8 is in Appendix F. Lemma 8 characterizes the posterior of the global latent variables $\theta$ when we perform variational approximation on the local latent variables $z$ under model misspecification.

Building on Lemma 8, Lemma 9 and Lemma 10 characterize the KL minimizer to the VB ideal within the mean field (i.e. factorizable) variational family $\mathcal{Q}^d$. They pave the road for characterizing the VB posterior of the global latent variables $\theta$.

**Lemma 9.** *The* KL *minimizer of the* VB *ideal over the mean field family is consistent: almost surely under* $P_0$, *it converges to a point mass centered at* $\theta^*$,

$$\underset{q(\theta) \in \mathcal{Q}^d}{\arg\min} \mathrm{KL}(q(\theta)||\pi^*(\theta \mid x)) \xrightarrow{d} \delta_{\theta^*}.$$

The proof of Lemma 9 is in Appendix G. Lemma 9 shows that the KL minimizer to the VB ideal converges to a point mass centered at $\theta^*$. Lemma 9 is intuitive in that (1) the VB ideal converges to a point mass centered at $\theta^*$ and (2) the point mass $\delta_{\theta^*}$ resides in the variational family $\mathcal{Q}^d$.

**Lemma 10.** *The* KL *minimizer of the* VB *ideal of* $\tilde{\theta}$ *converges to that of* $\mathcal{N}(\cdot; \Delta_{n,\theta^*}, V_{\theta^*}^{-1})$ *in total variation: under mild technical conditions on the tail behavior of* $\mathcal{Q}^d$ *(see Assumption 7 in Appendix H),*

$$\left\| \underset{q \in \mathcal{Q}^d}{\arg\min} \mathrm{KL}(q(\cdot)||\pi_{\tilde{\theta}}^*(\cdot \mid x)) - \underset{q \in \mathcal{Q}^d}{\arg\min} \mathrm{KL}(q(\cdot)||\mathcal{N}(\cdot; \Delta_{n,\theta^*}, V_{\theta^*}^{-1})) \right\|_{\mathrm{TV}} \xrightarrow{P_0} 0.$$

The proof of Lemma 10 is in Appendix H. Lemma 10 shows that the KL minimizer to the VB ideal converges to the KL minimizer to $\mathcal{N}(\cdot; \Delta_{n,\theta^*}, V_{\theta^*}^{-1})$. As with Lemma 9, Lemma 10 is also intuitive because Lemma 8 has shown that $\pi_{\tilde{\theta}}^*(\cdot \mid x)$ and $\mathcal{N}(\cdot; \Delta_{n,\theta^*}, V_{\theta^*}^{-1})$ are close in the large sample limit.

The final step of the proof is to establish the connection between the VB posterior and the KL minimizer of the VB ideal. First notice that the VB posterior is then the minimizer of the so-called profiled ELBO:

$$q^*(\theta) = \underset{q(\theta)}{\arg\max} \mathrm{ELBO}_p(q(\theta)), \tag{31}$$

which treats the variational posterior of local latent variables $z$'s as a function of $q(\theta)$. Technically, the profiled ELBO is defined as follows:

$$\mathrm{ELBO}_p(q(\theta)) := \underset{q(z)}{\sup} \int q(\theta) \left( \log \left[ p(\theta) \exp \left\{ \int q(z) \log \frac{p(x, z \mid \theta)}{q(z)} \, \mathrm{d}z \right\} \right] - \log q(\theta) \right) \mathrm{d}\theta. \tag{32}$$

Via this representation of the VB posterior, Lemma 4 of Wang & Blei [11] shows that the VB posterior and the KL minimizer of the VB ideal are close in the large sample limit. We restate this result here for completeness.

**Lemma 11** (Lemma 4 of Wang & Blei [11])**.** *The negative* KL *divergence to the* VB *ideal is equivalent to the profiled* ELBO *in the limit: under mild technical conditions on the tail behavior of* $\mathcal{Q}^d$, *for* $q(\theta) \in \mathcal{Q}^d$,

$$\mathrm{ELBO}_p(q(\theta)) = -\mathrm{KL}(q(\theta)||\pi^*(\theta \mid x)) + o_{P_0}(1).$$

Given Lemmas 8 to 11, we can prove Theorem 3.

Theorem 3.1 and Theorem 3.2 are direct consequences of Lemmas 9 to 11. We have

$$\left\| \underset{q \in \mathcal{Q}^d}{\arg\max} \, \mathrm{ELBO}_p(q(\tilde{\theta})) - \underset{q \in \mathcal{Q}^d}{\arg\min} \, \mathrm{KL}(q(\cdot) \| \mathcal{N}(\cdot \, ; \Delta_{n,\theta^*}, V_{\theta^*}^{-1})) \right\|_{\mathrm{TV}} \overset{P_0}{\to} 0,$$

which leads to the consistency and asymptotic normality of $q^*(\theta)$ due to Eq. 31.

Theorem 3.3 and Theorem 3.4 follows from Lemmas 9 to 11 via a similar proof argument with Theorem 2.3 in Kleijn et al. [6] and Theorem 10.8 in Van der Vaart [10].

We consider three stochastic processes: fix some compact set $K$ and for given $M > 0$,

$$t \mapsto Z_{n,M}(t) = \int_{\|\tilde{\theta}\| \leq M} (t - \tilde{\theta})^2 \cdot q_{\tilde{\theta}}^*(\tilde{\theta}) \, \mathrm{d}\tilde{\theta}, \tag{33}$$

$$t \mapsto W_{n,M}(t) = \int_{\|\tilde{\theta}\| \leq M} (t - \tilde{\theta})^2 \cdot \mathcal{N}(\tilde{\theta}; \Delta_{n,\theta^*}, (V'_{\theta^*})^{-1}) \, \mathrm{d}\tilde{\theta}, \tag{34}$$

$$t \mapsto W_M(t) = \int_{\|\tilde{\theta}\| \leq M} (t - \tilde{\theta})^2 \cdot \mathcal{N}(\tilde{\theta}; X, (V'_{\theta^*})^{-1}) \, \mathrm{d}\tilde{\theta}. \tag{35}$$

The intuition behind these constructions is that (1) $\tilde{\theta}^{\mathrm{VB}} = \delta_n^{-1}(\hat{\theta}_{\mathrm{VB}} - \theta^*)$ is the minimizer of the process $t \mapsto Z_{n,\infty}(t)$ and (2) $X = \int \tilde{\theta} \cdot \mathcal{N}(\tilde{\theta}; X, V_{\theta^*}'^{-1}) \, \mathrm{d}\tilde{\theta}$ is the minimizer of $t \mapsto W_\infty(t)$.

To prove Theorem 3.3 and Theorem 3.4, we have

$$Z_{n,M} - W_{n,M} = o_{P_0}(1) \text{ in } \ell^\infty(K)$$

due to Theorem 3.2 and $\sup_{t \in K, \|h\| \leq M} (t - h)^2 < \infty$. Then we have

$$W_{n,M} - W_M = o_{P_0}(1) \text{ in } \ell^\infty(K)$$

due to $\Delta_{n,\theta^*} \overset{d}{\to} X$ and the continuous mapping theorem. Finally, we have

$$W_M - W_\infty = o_{P_0}(1) \text{ in } \ell^\infty(K)$$

as $M \to \infty$ because of $\int \theta \cdot q^*(\theta) < \infty$, and

$$Z_{n,p^{\mathrm{VB}}} - Z_{n,\infty} = o_{P_0}(1) \text{ in } \ell^\infty(K)$$

due to $\int_{\|\tilde{\theta}\| > p^{\mathrm{VB}}} \|\tilde{\theta}\|^2 q^*(\tilde{\theta}) \, \mathrm{d}\tilde{\theta} \overset{P_0}{\to} 0$ for for any $p^{\mathrm{VB}} \to \infty$ ensured by Assumption 7.1. Therefore, we have

$$Z_{n,\infty} - W_\infty = o_{P_0}(1) \text{ in } \ell^\infty(K),$$

which implies

$$\tilde{\theta}^{\mathrm{VB}} \overset{d}{\to} X$$

due to the continuity and convexity of the squared loss and the argmax theorem.

$\square$

# E   Proof of Theorems 2 and 4

We prove Theorem 4 here. Theorem 2 is a direct consequence of Theorem 4.

*Proof.* We next study the posterior predictive distribution resulting from the VB posterior. For notational simplicity, we abbreviate $p_{\mathrm{VB}}^{\mathrm{pred}}(x_{\mathrm{new}} \mid x_{1:n})$ as $p_{\mathrm{VB}}^{\mathrm{pred}}(x_{\mathrm{new}})$.

$$\left\| p_{\text{VB}}^{\text{pred}}(x_{\text{new}}) - p_{\text{true}}^{\text{pred}}(x_{\text{new}}) \right\|_{\text{TV}} \tag{36}$$

$$= \left\| \int p(x_{\text{new}} \mid \theta) q^*(\theta) \, d\theta - \int p(x_{\text{new}} \mid \theta) p(\theta \mid x) \, d\theta \right\|_{\text{TV}} \tag{37}$$

$$= \frac{1}{2} \int \left| \int p(x_{\text{new}} \mid \theta) q^*(\theta) \, d\theta - \int p(x_{\text{new}} \mid \theta) p(\theta \mid x) \, d\theta \right| dx_{\text{new}} \tag{38}$$

$$= \frac{1}{2} \int \left| \int p(x_{\text{new}} \mid \theta) \left( q^*(\theta) - p(\theta \mid x) \right) d\theta \right| dx_{\text{new}} \tag{39}$$

$$\leq \frac{1}{2} \int \left| \int p(x_{\text{new}} \mid \theta^*) \left( q^*(\theta) - p(\theta \mid x) \right) d\theta \right| dx_{\text{new}} \tag{40}$$

$$+ \frac{1}{2} \int \left| \int \left( p(x_{\text{new}} \mid \theta) - p(x_{\text{new}} \mid \theta^*) \right) \cdot \left( q^*(\theta) - p(\theta \mid x) \right) d\theta \right| dx_{\text{new}} \tag{41}$$

The first equality is due to the definition of posterior predictive densities. The second equality is due to the definition of the total variation (TV) distance. The third equality collects the two integrals into one. The fourth equality is due to $p_\theta(x_{\text{new}}) \geq 0$ and triangle inequality.

If each term in Eq. 41 goes to zero in the large sample limit, we have

$$\left\| p_{\text{VB}}^{\text{pred}}(x_{\text{new}}) - p_{\text{true}}^{\text{pred}}(x_{\text{new}}) \right\|_{\text{TV}} \xrightarrow{P_0} 0. \tag{42}$$

Moreover, we assume the model $\{p_\theta : \theta \in \Theta\}$ is misspecified, which implies

$$\left\| p_0(x_{\text{new}}) - p_{\text{true}}^{\text{pred}}(x_{\text{new}}) \right\|_{\text{TV}} \tag{43}$$

$$\geq \| p_0(x_{\text{new}}) - p(x_{\text{new}} \mid \theta^*) \|_{\text{TV}} - \left\| p_{\text{true}}^{\text{pred}}(x_{\text{new}}) - p(x_{\text{new}} \mid \theta^*) \right\|_{\text{TV}} \tag{44}$$

$$\xrightarrow{P_0} \| p_0(x_{\text{new}}) - p(x_{\text{new}} \mid \theta^*) \|_{\text{TV}} \tag{45}$$

$$> 0. \tag{46}$$

The first inequality is due to triangle inequality. The second equation is due to a similar argument with $\left\| p_{\text{VB}}^{\text{pred}}(x_{\text{new}}) - p_{\text{true}}^{\text{pred}}(x_{\text{new}}) \right\|_{\text{TV}} \xrightarrow{P_0} 0$. The intuition is that the posterior $p(\theta \mid x) \xrightarrow{P_0} \delta_{\theta^*}$ in the large sample limit, so the posterior predictive distribution should converge to $p(x_{\text{new}} \mid \theta^*) = \int p_\theta(x_{\text{new}}) \delta_{\theta^*}(\theta) \, d\theta$. The last step $\| p_0(x_{\text{new}}) - p(x_{\text{new}} \mid \theta^*) \|_{\text{TV}} > 0$ is due to the assumption that the model $p(\cdot \mid \theta)$ is misspecified.

Eq. 42 and Eq. 43 together imply Theorem 4:

$$\frac{\left\| p_{\text{VB}}^{\text{pred}}(x_{\text{new}}) - p_{\text{true}}^{\text{pred}}(x_{\text{new}}) \right\|_{\text{TV}}}{\left\| p_0(x_{\text{new}}) - p_{\text{true}}^{\text{pred}}(x_{\text{new}}) \right\|_{\text{TV}}} \xrightarrow{P_0} 0.$$

Below we show that each term in Eq. 41 goes to zero in the large sample limit, which completes the proof.

For the first term in Eq. 41, we have.

$$\int \left| \int p(x_{\text{new}} \mid \theta^*) \left( q^*(\theta) - p(\theta \mid x) \right) d\theta \right| dx_{\text{new}}$$

$$= \int \left| p(x_{\text{new}} \mid \theta^*) \int \left( q^*(\theta) - p(\theta \mid x) \right) d\theta \right| dx_{\text{new}}$$

$$= \int | p(x_{\text{new}} \mid \theta^*) \cdot 0 | \, dx_{\text{new}}$$

$$= 0.$$

The first equality is due to $p(x_{\text{new}} \,|\, \theta^*)$ not depending on $\theta$. The second equality is due to both $q^*(\theta)$ and $p(\theta \,|\, x)$ being probability density functions. The third equality is due to integration of zero equal to zero.

Next we note that

$$
\int \left| \int \left( p(x_{\text{new}} \,|\, \theta) - p(x_{\text{new}} \,|\, \theta^*) \right) \cdot \left( q^*(\theta) - p(\theta \,|\, x) \right) \mathrm{d}\theta \right| \mathrm{d}x_{\text{new}}
$$
$$
\leq \int \left| \int \left( p(x_{\text{new}} \,|\, \theta) - p(x_{\text{new}} \,|\, \theta^*) \right) \cdot \left( q^*(\theta) - \mathcal{N}(\theta \,;\, \theta^*, \delta_n^\top V_{\theta^*}^{'-1} \delta_n) \right) \mathrm{d}\theta \right| \mathrm{d}x_{\text{new}}
$$
$$
+ \int \left| \int \left( p(x_{\text{new}} \,|\, \theta) - p(x_{\text{new}} \,|\, \theta^*) \right) \cdot \left( \mathcal{N}(\theta \,;\, \theta^*, \delta_n^\top V_{\theta^*}^{'-1} \delta_n) - \mathcal{N}(\theta \,;\, \theta^*, \delta_n^\top V_{\theta^*}^{-1} \delta_n) \right) \mathrm{d}\theta \right| \mathrm{d}x_{\text{new}}
$$
$$
+ \int \left| \int \left( p(x_{\text{new}} \,|\, \theta) - p(x_{\text{new}} \,|\, \theta^*) \right) \cdot \left( \mathcal{N}(\theta \,;\, \theta^*, \delta_n^\top V_{\theta^*}^{-1} \delta_n) - p(\theta \,|\, x) \right) \mathrm{d}\theta \right| \mathrm{d}x_{\text{new}}.
$$

We apply the Taylor's theorem to $p(x_{\text{new}} \,|\, \theta) - p(x_{\text{new}} \,|\, \theta^*)$: There exists some function $h_{\theta^*}(\theta)$ such that

$$
p(x_{\text{new}} \,|\, \theta) - p(x_{\text{new}} \,|\, \theta^*)
$$
$$
= (\theta - \theta^*) \cdot \nabla_\theta p(x_{\text{new}} \,|\, \theta) \big|_{\theta = \theta^*}
$$
$$
+ \nabla_\theta^2 p(x_{\text{new}} \,|\, \theta) \big|_{\theta = \theta^*} \cdot (\theta - \theta^*)(\theta - \theta^*)^\top
$$
$$
+ h_{\theta^*}(\theta) \cdot (\theta - \theta^*)(\theta - \theta^*)^\top,
$$

where $\lim_{\theta \to \theta^*} h_{\theta^*}(\theta) = 0$. We apply this expansion to each of the term above:

$$
\int \left| \int \left( p(x_{\text{new}} \,|\, \theta) - p(x_{\text{new}} \,|\, \theta^*) \right) \cdot \left( q^*(\theta) - \mathcal{N}(\theta \,;\, \theta^*, \delta_n^\top V_{\theta^*}^{'-1} \delta_n) \right) \mathrm{d}\theta \right| \mathrm{d}x_{\text{new}}
$$
$$
\leq \int \left| \int \left( (\theta - \theta^*) \cdot \nabla_\theta p(x_{\text{new}} \,|\, \theta) \big|_{\theta = \theta^*} \right) \cdot \left( q^*(\theta) - \mathcal{N}(\theta \,;\, \theta^*, \delta_n^\top V_{\theta^*}^{'-1} \delta_n) \right) \mathrm{d}\theta \right| \mathrm{d}x_{\text{new}}
$$
$$
+ \int \left| \int \left( \nabla_\theta^2 p(x_{\text{new}} \,|\, \theta) \big|_{\theta = \theta^*} \cdot (\theta - \theta^*)(\theta - \theta^*)^\top \right) \cdot \left( q^*(\theta) - \mathcal{N}(\theta \,;\, \theta^*, \delta_n^\top V_{\theta^*}^{'-1} \delta_n) \right) \mathrm{d}\theta \right| \mathrm{d}x_{\text{new}}
$$
$$
+ \int \left| \int \left( h_{\theta^*}(\theta) \cdot (\theta - \theta^*)(\theta - \theta^*)^\top \right) \cdot \left( q^*(\theta) - \mathcal{N}(\theta \,;\, \theta^*, \delta_n^\top V_{\theta^*}^{'-1} \delta_n) \right) \mathrm{d}\theta \right| \mathrm{d}x_{\text{new}}
$$
$$
\to 0 \cdot \int \left| \nabla_\theta p(x_{\text{new}} \,|\, \theta) \big|_{\theta = \theta^*} \right| \mathrm{d}x_{\text{new}} + 0 \cdot \int \left| \nabla_\theta^2 p(x_{\text{new}} \,|\, \theta) \big|_{\theta = \theta^*} \right| \mathrm{d}x_{\text{new}} + 0 \cdot \int |h_{\theta^*}(\theta)| \, \mathrm{d}x_{\text{new}}
$$
$$
= 0
$$

The key property that enables the calculation above is that $q^*(\theta)$ and $\mathcal{N}(\theta \,;\, \theta^*, \delta_n^\top V_{\theta^*}^{'-1} \delta_n)$ share the same first and second moments.

With the same argument, we can show that

$$
\int \left| \int \left( p(x_{\text{new}} \,|\, \theta) - p(x_{\text{new}} \,|\, \theta^*) \right) \cdot \left( \mathcal{N}(\theta \,;\, \theta^*, \delta_n^\top V_{\theta^*}^{-1} \delta_n) - p(\theta \,|\, x) \right) \mathrm{d}\theta \right| \mathrm{d}x_{\text{new}} \to 0.
$$

Finally, we work with the middle term.

$$\int \left| \int \left( p(x_{\text{new}} \mid \theta) - p(x_{\text{new}} \mid \theta^*) \right) \cdot \left( \mathcal{N}(\theta \,;\, \theta^*, \delta_n^\top V_{\theta^*}^{'-1} \delta_n) - \mathcal{N}(\theta \,;\, \theta^*, \delta_n^\top V_{\theta^*}^{-1} \delta_n) \right) \mathrm{d}\theta \right| \mathrm{d}x_{\text{new}}$$

$$\leq \int \left| \int \left( (\theta - \theta^*) \cdot \nabla_\theta p(x_{\text{new}} \mid \theta)\big|_{\theta = \theta^*} \right) \cdot \left( \mathcal{N}(\theta \,;\, \theta^*, \delta_n^\top V_{\theta^*}^{'-1} \delta_n) - \mathcal{N}(\theta \,;\, \theta^*, \delta_n^\top V_{\theta^*}^{-1} \delta_n) \right) \mathrm{d}\theta \right| \mathrm{d}x_{\text{new}}$$

$$+ \int \left| \int \left( \nabla_\theta^2 p(x_{\text{new}} \mid \theta)\big|_{\theta = \theta^*} \cdot (\theta - \theta^*)(\theta - \theta^*)^\top \right) \right.$$
$$\left. \cdot \left( \mathcal{N}(\theta \,;\, \theta^*, \delta_n^\top V_{\theta^*}^{'-1} \delta_n) - \mathcal{N}(\theta \,;\, \theta^*, \delta_n^\top V_{\theta^*}^{-1} \delta_n) \right) \mathrm{d}\theta \right| \mathrm{d}x_{\text{new}}$$

$$+ \int \left| \int \left( h_{\theta^*}(\theta) \cdot (\theta - \theta^*)(\theta - \theta^*)^\top \right) \cdot \left( \mathcal{N}(\theta \,;\, \theta^*, \delta_n^\top V_{\theta^*}^{'-1} \delta_n) - \mathcal{N}(\theta \,;\, \theta^*, \delta_n^\top V_{\theta^*}^{-1} \delta_n) \right) \mathrm{d}\theta \right| \mathrm{d}x_{\text{new}}$$

$$= 0 \cdot \int \left| \nabla_\theta p(x_{\text{new}} \mid \theta)\big|_{\theta = \theta^*} \right| \mathrm{d}x_{\text{new}}$$

$$+ (\delta_n^\top V_{\theta^*}^{-1} \delta_n - \delta_n^\top V_{\theta^*}^{'-1} \delta_n) \cdot \int \left| \nabla_\theta^2 p(x_{\text{new}} \mid \theta)\big|_{\theta = \theta^*} \right| \mathrm{d}x_{\text{new}}$$

$$+ (\delta_n^\top V_{\theta^*}^{-1} \delta_n - \delta_n^\top V_{\theta^*}^{'-1} \delta_n) \cdot \int \left| h_{\theta^*}(\theta) \right| \mathrm{d}x_{\text{new}}$$

$$\to 0$$

The last step is because $\delta_n \to 0$ and $\int \nabla_\theta^2 p(x_{\text{new}} \mid \theta)\big|_{\theta = \theta^*} \mathrm{d}x_{\text{new}} < \infty$.

$\square$

## F    Proof of Lemma 8

In this proof, we only need to show that Assumption 4 implies Assumption (2.3) in Kleijn et al. [6]: $\int_{\tilde{\theta} > p^{\text{VB}}} \pi_{\tilde{\theta}}^*(\tilde{\theta} \mid x) \, \mathrm{d}\tilde{\theta} \xrightarrow{P_0} 0$ for every sequence of constants $p^{\text{VB}} \to \infty$, where $\tilde{\theta} = \delta_n^{-1}(\theta - \theta^*)$.

To prove this implication, we repeat Theorem 3.1, Theorem 3.3, Lemma 3.3, Lemma 3.4 of Kleijn et al. [6]. The only difference is that we prove it for the general convergence $\delta_n$ instead of the parametric convergence rate $\sqrt{n}$. The idea is to consider test sequences of uniform exponential power around $\theta^*$. We omit the proof here; see Kleijn et al. [6] for details.

This proof also resembles the proof of Lemma 1 in Wang & Blei [11].

## G    Proof of Lemma 9

We focus on the VB posterior of $\theta$ which converges with the $\delta_n$ rate. Without loss of generality, we consider the subset of mean field variational family that also shrinks with the rate $\delta_n$. The rationale of this step is that the KL divergence between exact posterior and the VB posterior will blow up to $\infty$ for other classes of variational families. More precisely, we assume the following variational family $\mathcal{Q}$

$$q_{\check{\theta}}(\check{\theta}) = q(\mu + \delta_n \check{\theta}) |\det(\delta_n)|, \tag{47}$$

where $\check{\theta} := \delta_n^{-1}(\theta - \mu)$, for some $\mu \in \Theta$.

Note the variational family is allowed to center at any value, not necessarily at $\theta^*$.

We now characterize the limiting distribution of the KL minimizer of the VB ideal. In other words, the mass of the KL minimizer concentrates near $\theta^*$ as $n \to \infty$:

$$q^{\ddagger}(\theta) := \underset{q(\theta) \in \mathcal{Q}^d}{\arg\min} \, \mathrm{KL}(q(\theta) || \pi^*(\theta \mid x)) \xrightarrow{d} \delta_{\theta^*}.$$

It suffices to show $\int_{B(\theta^*, \xi_n)} q^{\ddagger}(\theta) \, \mathrm{d}\theta \xrightarrow{P_0} 1$, for some $\xi_n \to 0$ as $n \to \infty$ due to the Slutsky's theorem.

The proof below mimics the proof of Lemma 2 of [11] (also the Step 2 in the proof of Lemma 3.6 along with Lemma 3.7 in Lu et al. [7]) except we take care of the extra technicality due to model misspecification. We include the proof for completeness here.

We start with two claims that we will prove later.

$$\limsup_{n\to\infty} \min \mathrm{KL}(q(\theta)||\pi^*(\theta \mid x)) \le M, \tag{48}$$

$$\int_{\mathbb{R}^d \setminus K} q^{\ddagger}(\theta)\, \mathrm{d}\theta \to 0, \tag{49}$$

where $M > 0$ is some constant and $K$ is the compact set assumed in the local asymptotic normality condition. We will use them to upper bound and lower bound $\int_{B(\theta^*,\xi_n)} q^{\ddagger,K}(\theta)\, \mathrm{d}\theta$.

The upper bound of $\int_{B(\theta^*,\xi_n)} q^{\ddagger,K}(\theta)\, \mathrm{d}\theta$ is due to the LAN condition,

$$\int q^{\ddagger,K}(\theta) p^{\mathrm{VB}}(\theta\,;\,x)\, \mathrm{d}\theta$$

$$= \int q^{\ddagger,K}(\theta) \left[ p^{\mathrm{VB}}(\theta^*\,;\,x) + \delta_n^{-1}(\theta - \theta^*)^\top V_{\theta^*} \Delta_{n,\theta^*} \right.$$

$$\left. - \frac{1}{2}[\delta_n^{-1}(\theta - \theta^*)]^\top V_{\theta^*}[\delta_n^{-1}(\theta - \theta^*)] + o_P(1) \right] \mathrm{d}\theta$$

$$\le p^{\mathrm{VB}}(\theta^*\,;\,x) - C_1 \sum_{i=1}^d \frac{\eta^2}{\delta_{n,ii}^2} \int_{B(\theta^*,\eta)^c} q^{\ddagger,K}(\theta)\, \mathrm{d}\theta + o_P(1),$$

for large enough $n$ and $\eta << 1$ and some constant $C_1 > 0$.

The lower bound of the integral is due to the first claim:

$$\int q^{\ddagger,K}(\theta) p^{\mathrm{VB}}(\theta\,;\,x)\, \mathrm{d}\theta \ge p^{\mathrm{VB}}(\theta^*\,;\,x) - M_0, \tag{50}$$

for some large constant $M_0 > M$. This is due to two steps. First, Eq. 31 of Wang & Blei [11] gives

$$\mathrm{KL}(q^{\ddagger,K}(\theta)||\pi^*(\theta \mid x)) \tag{51}$$

$$= \log|\det(\delta_n)|^{-1} + \sum_{i=1}^d \mathbb{H}(q_{h,i}^{\ddagger,K}(h)) - \int q^{\ddagger,K}(\theta) \log \pi^*(\theta \mid x)\, \mathrm{d}\theta. \tag{52}$$

Then we approximate the last term by the LAN condition:

$$\int q^{\ddagger,K}(\theta) \log \pi^*(\theta \mid x)\, \mathrm{d}\theta \tag{53}$$

$$= \int q^{\ddagger,K}(\theta) \log p(\theta)\, \mathrm{d}\theta + \int q(\theta) p^{\mathrm{VB}}(\theta\,;\,x)\, \mathrm{d}\theta - \log \int p(\theta) \exp(p^{\mathrm{VB}}(\theta\,;\,x))\, \mathrm{d}\theta \tag{54}$$

$$= \int q^{\ddagger,K}(\theta) \log p(\theta)\, \mathrm{d}\theta + \int q^{\ddagger,K}(\theta) p^{\mathrm{VB}}(\theta\,;\,x)\, \mathrm{d}\theta$$

$$- \left[ \frac{d}{2}\log(2\pi) - \frac{1}{2}\log\det V_{\theta^*} + \log\det(\delta_n) + p^{\mathrm{VB}}(\theta^*\,;\,x) + \log p(\theta^*) + o_P(1) \right]. \tag{55}$$

The above approximation leads to the following approximation to the KL divergence:

$$\mathrm{KL}(q^{\ddagger,K}(\theta)||\pi^*(\theta \mid x)) \tag{56}$$

$$= \log|\det(\delta_n)|^{-1} + \sum_{i=1}^d \mathbb{H}(q_{h,i}^{\ddagger,K}(h)) - \int q^{\ddagger,K}(\theta) \log p(\theta)\, \mathrm{d}\theta - \int q^{\ddagger,K}(\theta) p^{\mathrm{VB}}(\theta\,;\,x)\, \mathrm{d}\theta$$

$$+ \left[ \frac{d}{2}\log(2\pi) - \frac{1}{2}\log\det V_{\theta^*} + \log\det(\delta_n) + p^{\mathrm{VB}}(\theta^*\,;\,x) + \log p(\theta^*) + o_P(1) \right] \tag{57}$$

$$= \sum_{i=1}^d \mathbb{H}(q_{h,i}^{\ddagger,K}(h)) - \int q^{\ddagger,K}(\theta) \log p(\theta)\, \mathrm{d}\theta - \int q^{\ddagger,K}(\theta) p^{\mathrm{VB}}(\theta\,;\,x)\, \mathrm{d}\theta$$

$$+ \frac{d}{2}\log(2\pi) - \frac{1}{2}\log\det V_{\theta^*} + p^{\mathrm{VB}}(\theta^*\,;\,x) + \log p(\theta^*) + o_P(1). \tag{58}$$

Then via the first claim above, we have

$$\int q^{\ddagger,K}(\theta)p^{\mathrm{VB}}(\theta\,;\,x)\,\mathrm{d}\theta \tag{59}$$

$$\geq -M + \sum_{i=1}^{d}\mathbb{H}(q_{h,i}^{\ddagger,K}(h)) - \int q^{\ddagger,K}(\theta)\log p(\theta)\,\mathrm{d}\theta$$

$$+\frac{d}{2}\log(2\pi) - \frac{1}{2}\log\det V_{\theta^*} + p^{\mathrm{VB}}(\theta^*\,;\,x) + \log p(\theta^*) + o_P(1) \tag{60}$$

$$\geq -M_0 + p^{\mathrm{VB}}(\theta^*\,;\,x) + o_P(1) \tag{61}$$

for some constant $M_0 > 0$. The last step is because the only term that depends on $n$ is $\int q^{\ddagger,K}(\theta)\log p(\theta)\,\mathrm{d}\theta$ which is finite due to Assumption 1.

Combining the lower and upper bounds of the integral gives

$$p^{\mathrm{VB}}(\theta^*\,;\,x) - C_1\sum_{i=1}^{d}\frac{\eta^2}{\delta_{n,ii}^2}\int_{B(\theta^*,\eta)^c}q^{\ddagger,K}(\theta)\,\mathrm{d}\theta + o_P(1) \geq \quad -M_0 + p^{\mathrm{VB}}(\theta^*\,;\,x)$$

$$\Rightarrow \int_{B(\theta^*,\eta)^c}q^{\ddagger,K}(\theta)\,\mathrm{d}\theta + o_P(1) \leq \frac{M_0\cdot(\min_i\delta_{n,ii})^2}{C_2\eta^2},$$

for some constant $C_2 > 0$. By choosing $\eta = \sqrt{M_0(\min_i\delta_{n,ii})/C_2} \to 0$, this term go to zero as $n$ goes to infinity. In other words, we have shown $\int_{B(\theta^*,\xi_n)}q^{\ddagger}(\theta)\,\mathrm{d}\theta \xrightarrow{P_0} 1$ with $\xi_n = \eta$.

We now prove the two claims made at the beginning. To show Eq. 48, it suffices to show that there exists a choice of $q(\theta)$ such that

$$\limsup_{n\to\infty}\mathrm{KL}(q(\theta)||\pi^*(\theta\mid x)) < \infty.$$

We choose $\tilde{q}(\theta) = \prod_{i=1}^{d}N(\theta_i;\theta_{0,i},\delta_{n,ii}^2 v_i)$ for $v_i > 0, i = 1,...,d$. We thus have

$$\mathrm{KL}(\tilde{q}(\theta)||\pi^*(\theta\mid x)) \tag{62}$$

$$= \sum_{i=1}^{d}\frac{1}{2}\log(v_i) + \frac{d}{2} + d\log(2\pi) - \int\tilde{q}(\theta)\log p(\theta)\,\mathrm{d}\theta - \int\tilde{q}(\theta)p^{\mathrm{VB}}(\theta\,;\,x)\,\mathrm{d}\theta$$

$$-\frac{1}{2}\log\det V_{\theta^*} + p^{\mathrm{VB}}(\theta^*\,;\,x) + \log p(\theta^*) + o_P(1) \tag{63}$$

$$= \sum_{i=1}^{d}\frac{1}{2}\log(v_i) + \frac{d}{2} + d\log(2\pi) - \frac{1}{2}\log\det V_{\theta^*} + C_6 + o_P(1), \tag{64}$$

for some constant $C_6 > 0$. The finiteness of limsup is due to the boundedness of the last term. The second equality is due to the limit of $\tilde{q}(\theta)$ concentrating around $\theta^*$. Specifically, we expand $\log p(\theta)$ to the second order around $\theta^*$,

$$\int\tilde{q}(\theta)\log p(\theta)\,\mathrm{d}\theta$$

$$= \log p(\theta^*) + \int\tilde{q}(\theta)\left[(\theta - \theta^*)(\log p(\theta^*))' + \frac{(\theta-\theta^*)^2}{2}\int_0^1(\log p(\xi\theta + (1-\xi)\theta^*))''(1-\xi)^2\,\mathrm{d}\xi\right]\mathrm{d}\theta$$

$$\leq \log p(\theta^*) + \frac{1}{2!}\max_{\xi\in[0,1]}\left\{\int\tilde{q}(\theta)(\theta-\theta^*)^2(\log p(\xi\theta + (1-\xi)\theta^*))''\,\mathrm{d}\theta\right\}$$

$$\leq \log p(\theta^*) + \frac{M_p}{\sqrt{(2\pi)^d\det(\delta_n^2)\prod_i v_i}}\int_{\mathbb{R}^d}|\theta|^2 e^{(|\theta|+|\theta^*|)^2}\cdot e^{-\frac{1}{2}\theta^\top(\delta_n V\delta_n)^{-1}\theta}\,\mathrm{d}\theta$$

$$\leq \log p(\theta^*) + \frac{M_p}{\sqrt{(2\pi)^d\det(\delta_n^2)\prod_i v_i}}e^{\theta^{*2}}\int_{\mathbb{R}^d}|\theta|^2 e^{-\frac{1}{2}\theta^\top[(\delta_n V\delta_n)^{-1}-2I_d]\theta}$$

$$\leq \log p(\theta^*) + C_3 M_p e^{\theta^{*2}}\max_d(\delta_{n,ii}^2)\det(V^{-1}-2\delta_n^2)^{-1}$$

$$\leq \log p(\theta^*) + C_4\max_d(\delta_{n,ii}^2)$$

where $\max_d(\delta_{n,ii}^2) \to 0$ and $C_3, C_4 > 0$. The first two equalities are due to Taylor expansion. The third inequality is due to the tail condition in Assumption 1. The fourth and fifth are due to rescaling $\theta$ appealing to the mean of a Chi-squared distribution with $d$ degrees of freedom. The last inequality is due to $\det(V^{-1} - 2\delta_n^2)^{-1} > 0$ for large enough $n$.

We apply the same Taylor expansion argument to the $\int \tilde{q}(\theta) p^{\mathrm{VB}}(\theta\,;\,x)\,\mathrm{d}\theta$ leveraging the LAN condition

$$\int_{K_n} \tilde{q}(\theta) p^{\mathrm{VB}}(\theta\,;\,x)\,\mathrm{d}\theta$$

$$= p^{\mathrm{VB}}(\theta^*\,;\,x) + \int_{K_n} \tilde{q}(\theta)\left[\delta_n^{-1}(\theta - \theta^*)^\top V_{\theta^*}\Delta_{n,\theta^*} + \frac{1}{2}(\delta_n^{-1}(\theta - \theta^*))^\top V_{\theta^*}\delta_n^{-1}(\theta - \theta^*) + o_P(1)\right]\mathrm{d}\theta$$

$$\leq p^{\mathrm{VB}}(\theta^*\,;\,x) + C_6 + o_P(1)$$

where $K_n$ is a compact set and $C_6 > 0$ is some constant.

For the set outside of this compact set $K_n$, choose $\tilde{q}(\theta) = \mathcal{N}(\theta; \theta^* + \Delta_{n,\theta^*}, \delta_n V_{\theta^*}\delta_n)$.

$$\int_{\mathbb{R}^d \setminus K_n} \tilde{q}(\theta)(\log p(\theta) + p^{\mathrm{VB}}(\theta\,;\,x))\,\mathrm{d}\theta \tag{65}$$

$$\leq C_7 \int_{\mathbb{R}^d \setminus K_n} \mathcal{N}(\theta; \theta^* + \Delta_{n,\theta^*}, \delta_n V_{\theta^*}\delta_n)(\log p(\theta) + p^{\mathrm{VB}}(\theta\,;\,x))\,\mathrm{d}\theta \tag{66}$$

$$\leq C_8[\det(\delta_n)^{-1}\log(\det(\delta_n)^{-1})]\int_{\mathbb{R}^d \setminus K_n} \mathcal{N}(\tilde{\theta}; \Delta_{n,\theta^*}, V_{\theta^*})\log \pi^*(\tilde{\theta} \mid x)\det(\delta_n)\,\mathrm{d}\tilde{\theta} \tag{67}$$

$$\leq C_9 \log(\det(\delta_n)^{-1})]\int_{\mathbb{R}^d \setminus K_n} [\pi^*(\tilde{\theta} \mid x) + o_P(1)]\log \pi^*(\tilde{\theta} \mid x), V_{\theta^*})\,\mathrm{d}\tilde{\theta} \tag{68}$$

$$\leq C_{10}\log(\det(\delta_n)^{-1})]\int_{\mathbb{R}^d \setminus K_n} [\mathcal{N}(\tilde{\theta}; \Delta_{n,\theta^*}, V_{\theta^*}) + o_P(1)]\log \mathcal{N}(\tilde{\theta}; \Delta_{n,\theta^*}, V_{\theta^*})\,\mathrm{d}\tilde{\theta} \tag{69}$$

$$\leq o_P(1) \tag{70}$$

for some $C_7, C_8, C_9, C_{10} > 0$. The first two inequalities are due to $\tilde{q}(\theta)$ centering at $\theta^*$ and a change of variable step. The third and fourth inequality is due to Lemma 8 and Theorem 2 in Piera & Parada [9]. The fifth inequality is due to a choice of fast enough increasing sequence of compact sets $K_n$.

We repeat this argument for the lower bound of $\int \tilde{q}(\theta)(\log p(\theta) + p^{\mathrm{VB}}(\theta\,;\,x))\,\mathrm{d}\theta$. Hence the first claim is proved.

To prove the second claim Eq. 49, we note that, for each $\epsilon > 0$, there exists an $N$ such that for all $n > N$ we have $\int_{||\theta - \mu|| > M} q(\theta)\,\mathrm{d}\theta < \epsilon$ because $\mathcal{Q}^d$ has a shrinking-to-zero scale. It leads to

$$\int_{\mathbb{R}^d \setminus K} q^\ddagger(\theta)\,\mathrm{d}\theta \leq \int_{\mathbb{R}^d \setminus B(\mu, M)} q^\ddagger(\theta)\,\mathrm{d}\theta \leq \epsilon.$$

# H  Proof of Lemma 10

*Proof.* To show the convergence of optimizers from two minimization problems, we invoke $\Gamma$-convergence: if two functionals $\Gamma-$converge, then their minimizer also converge. We refer the readers to Appendix C of Wang & Blei [11] for a review of $\Gamma$-convergence.

For notation convenience, we index the variational family by some finite dimensional parameter $m$. The goal is to show
$$F_n(m) := \mathrm{KL}(q(\theta; m) || \pi^*(\theta \mid x))$$
$\Gamma$-converges to
$$F_0(m) := \mathrm{KL}(q(\theta; m) || \mathcal{N}(\theta; \theta^* + \delta_n\Delta_{n,\theta^*}, \delta_n V_{\theta^*}^{-1}\delta_n)) - \Delta_{n,\theta^*}^\top V_{\theta^*}\Delta_{n,\theta^*}$$
in $P_0$-probability as $n \to 0$.

Write the densities in the mean field variational family in the following form: $q(\theta) = \prod_{i=1}^d \delta_{n,ii}^{-1} q_{h,i}(h)$, where $h = \delta_n^{-1}(\theta - \mu)$ for some $\mu \in \Theta$. This form of density is consistent with the change of variable step in Appendix G.

**Assumption 7.** *We assume the following conditions on $q_{h,i}$:*

1. *$q_{h,i}, i = 1, ..., d$ have continuous densities, positive and finite entropies, and $\int q'_{h,i}(h)\,\mathrm{d}h < \infty, i = 1, ..., d$.*

2. *If $q_h$ is has zero mean, we assume $\int h^2 \cdot q_h(h)\,\mathrm{d}h < \infty$ and $\sup_{z,x} |(\log p(z, x \mid \theta))''| \le C_{11} \cdot q_h(\theta)^{-C_{12}}$ for some $C_{11}, C_{12} > 0$; $|p^{\mathrm{VB}}(\theta\,;\, x)''| \le C_{13} \cdot q_h(\theta)^{-C_{14}}$ for some $C_{13}, C_{14} > 0$.*

3. *If $q_h$ has nonzero mean, we assume $\int h \cdot q_h(h)\,\mathrm{d}h < \infty$ and $\sup_{z,x} |(\log p(z, x \mid \theta))'| \le C_{11} \cdot q_h(\theta)^{-C_{12}}$ for some $C_{11}, C_{12} > 0$; $|p^{\mathrm{VB}}(\theta\,;\, x)'| \le C_{13}| \cdot q_h(\theta)^{-C_{14}}$ for some $C_{13}, C_{14} > 0$.*

Assumption 7.1 ensures that convergence in the finite-dimensional parameter implied convergence in TV distance due to Eqs 64-68 of Wang & Blei [11]. Assumption 7 is analogous to Assumptions 2 and 3 of Wang & Blei [11].

Leveraging the fundamental theorem of $\Gamma-$convergence [2, 3], the $\Gamma$-convergence of the two functionals implies $m_n \overset{P_0}{\to} m_0$; $m_n$ minimizes $F_n$ and $m_0$ minimizes $F_0$. Importantly, this is true because $\Delta_{n,\theta^*}^\top V_{\theta^*} \Delta_{n,\theta^*}$ is a constant bounded in $P_0$ probability and does not depend on $m$. The convergence in total variation then follows from Assumption 7.

Therefore, what remains is to prove the $\Gamma$-convergence of the two functionals.

We first rewrite $F_n(m, \mu)$.

$$F_n(m, \mu) := \mathrm{KL}(q(\theta; m, \mu) || \pi^*(\theta \mid x)) \tag{71}$$

$$= \log |\det(\delta_n)|^{-1} + \sum_{i=1}^d \mathbb{H}(q_{h,i}(h; m)) - \int q(\theta; m, \mu) \log p(\theta)\,\mathrm{d}\theta - \int q(\theta; m, \mu) p^{\mathrm{VB}}(\theta\,;\, x)\,\mathrm{d}\theta$$

$$+ \log \int p(\theta) \exp(p^{\mathrm{VB}}(\theta\,;\, x))\,\mathrm{d}\theta \tag{72}$$

$$= \log |\det(\delta_n)|^{-1} + \sum_{i=1}^d \mathbb{H}(q_{h,i}(h; m)) - \int q(\theta; m, \mu) \log p(\theta)\,\mathrm{d}\theta - \int q(\theta; m, \mu) p^{\mathrm{VB}}(\theta\,;\, x)\,\mathrm{d}\theta$$

$$+ \left[ \frac{d}{2} \log(2\pi) - \frac{1}{2} \log \det V_{\theta^*} + \log \det(\delta_n) + p^{\mathrm{VB}}(\theta^*\,;\, x) + \log p(\theta^*) + o_P(1) \right] \tag{73}$$

$$= \sum_{i=1}^d \mathbb{H}(q_{h,i}(h; m)) - \int q(\theta; m, \mu) p^{\mathrm{VB}}(\theta\,;\, x)\,\mathrm{d}\theta + \log p(\theta^*) - \log p(\mu)$$

$$+ \left[ \frac{d}{2} \log(2\pi) - \frac{1}{2} \log \det V_{\theta^*} + p^{\mathrm{VB}}(\theta^*\,;\, x) + o_P(1) \right] \tag{74}$$

$$= \sum_{i=1}^d \mathbb{H}(q_{h,i}(h; m)) - \int \delta_n^{-1}(\theta - \theta^*)^\top V_{\theta^*} \Delta_{n,\theta^*} \cdot q(\theta; m, \mu)\,\mathrm{d}\theta$$

$$+ \int \frac{1}{2} (\delta_n^{-1}(\theta - \theta^*))^\top V_{\theta^*} \delta_n^{-1}(\theta - \theta^*) \cdot q(\theta; m, \mu)\,\mathrm{d}\theta - \left[ \frac{d}{2} \log(2\pi) - \frac{1}{2} \log \det V_{\theta^*} + o_P(1) \right], \tag{75}$$

due to algebraic operations and the LAN condition of $p^{\mathrm{VB}}(\theta\,;\, x)$. To extend from the compact set $K$ to the whole space $\mathbb{R}^d$, we employ the same argument as in Eq. 64.

Next we rewrite $F_0(m, \mu)$.

$$\mathrm{KL}(q(\theta; m, \mu) || \mathcal{N}(\theta; \theta^* + \delta_n \Delta_{n, \theta^*}, \delta_n V_{\theta^*}^{-1} \delta_n))$$

$$= \log |\det(\delta_n)|^{-1} + \sum_{i=1}^{d} \mathbb{H}(q_{h,i}(h; m)) + \int q(\theta; m, \mu) \log \mathcal{N}(\theta; \theta^* + \delta_n \Delta_{n, \theta^*}, \delta_n V_{\theta^*}^{-1} \delta_n) \, \mathrm{d}\theta$$

$$= \log |\det(\delta_n)|^{-1} + \sum_{i=1}^{d} \mathbb{H}(q_{h,i}(h; m)) + \frac{d}{2} \log(2\pi) - \frac{1}{2} \log \det V_{\theta^*} + \log \det(\delta_n)$$

$$+ \int q(\theta; m, \mu) \cdot (\theta - \theta^* - \delta_n \Delta_{n, \theta^*})^\top \delta_n^{-1} V_{\theta^*} \delta_n^{-1} (\theta - \theta^* - \delta_n \Delta_{n, \theta^*}) \, \mathrm{d}\theta$$

$$= \sum_{i=1}^{d} \mathbb{H}(q_{h,i}(h; m)) + \frac{d}{2} \log(2\pi) - \frac{1}{2} \log \det V_{\theta^*} + \Delta_{n, \theta^*}^\top V_{\theta^*} \Delta_{n, \theta^*}$$

$$- \int \delta_n^{-1} (\theta - \theta^*)^\top V_{\theta^*} \Delta_{n, \theta^*} \cdot q(\theta; m, \mu) \, \mathrm{d}\theta + \int \frac{1}{2} (\delta_n^{-1} (\theta - \theta^*))^\top V_{\theta^*} \delta_n^{-1} (\theta - \theta^*) \cdot q(\theta; m, \mu) \, \mathrm{d}\theta.$$

These representations of $F_0(m, \mu)$ and $F_n(m, \mu)$ leads to

$$F_0(m, \mu) - \Delta_{n, \theta^*}^\top V_{\theta^*} \Delta_{n, \theta^*} \tag{76}$$

$$= \sum_{i=1}^{d} \mathbb{H}(q_{h,i}(h; m)) - \frac{d}{2} \log(2\pi) + \frac{1}{2} \log \det V_{\theta^*}$$

$$- \int \delta_n^{-1} (\theta - \theta^*)^\top V_{\theta^*} \Delta_{n, \theta^*} \cdot q(\theta; m, \mu) \, \mathrm{d}\theta$$

$$+ \int \frac{1}{2} (\delta_n^{-1} (\theta - \theta^*))^\top V_{\theta^*} \delta_n^{-1} (\theta - \theta^*) \cdot q(\theta; m, \mu) \, \mathrm{d}\theta \tag{77}$$

$$= +\infty \cdot (1 - \mathbb{I}_\mu(\theta^*)) + [\sum_{i=1}^{d} \mathbb{H}(q_{h,i}(h; m)) - \frac{d}{2} \log(2\pi) + \frac{1}{2} \log \det V_{\theta^*}$$

$$- \int \delta_n^{-1} (\theta - \theta^*)^\top V_{\theta^*} \Delta_{n, \theta^*} \cdot q(\theta; m, \mu) \, \mathrm{d}\theta$$

$$+ \int \frac{1}{2} (\delta_n^{-1} (\theta - \theta^*))^\top V_{\theta^*} \delta_n^{-1} (\theta - \theta^*) \cdot q(\theta; m, \mu) \, \mathrm{d}\theta] \cdot \mathbb{I}_\mu(\theta^*) \tag{78}$$

where the last equality is due to Assumption 7.

Comparing Eq. 75 and Eq. 78, we can prove the $\Gamma$ convergence.

Let $m_n \to m$. When $\mu \neq \theta^*$, $\liminf_{n \to \infty} F_n(m_n, \mu) = +\infty$. When $\mu = \theta^*$, we have $F_n(m, \mu) = F_0(m, \mu) - \Delta_{n, \theta^*}^\top V_{\theta^*} \Delta_{n, \theta^*} + o_P(1)$, which implies $F_0(m, \mu) \leq \lim_{n \to \infty} F_n(m_n, \mu)$ in $P_0$ probability by Assumption 7. These implies the limsup inequality required by $\Gamma$-convergence.

We then show the existence of a recovery sequence. When $\mu \neq \theta^*$, $F_0(m, \mu) = +\infty$. When $\mu = \theta^*$, we can simply choose $m_n = \theta^*$. Then we have $F_0(m, \mu) \leq \lim_{n \to \infty} F_n(p^{\mathrm{VB}}, \mu)$ in $P_0$ probability and the continuity of $F_n$. The $\Gamma$-convergence of the $F$ functionals then follows from the limsup inequalities above and the existence of recovery sequence.

Finally, we have

$$\arg\min F_0 = \arg\min F_0 - \Delta_{n, \theta^*}^\top V_{\theta^*} \Delta_{n, \theta^*}$$

because $\Delta_{n, \theta^*}^\top V_{\theta^*} \Delta_{n, \theta^*}$ does not depend on $m$ or $\mu$. The convergence of KL minimizers are proved.

$\square$

# I   Details of the Simulations

We follow the protocol as implemented in Stan. For HMC, we run four parallel chains and use 10,000 burn-in samples, and determine mixing using the R-hat convergence diagnostic (R-hat<1.01). For

variational Bayes, we run optimization until convergence (i.e. a local optimum). We cannot confirm if the local optimal we reached is global. Further, we conduct multiple parallel runs under each simulation setup and report the mean and the standard deviation of "RMSE" or "Mean KL." The error bars in Figure 2 are the standard deviation across different runs of the same simulation.