[Reviews · NeurIPS 2019]

Reviewer 1



he paper characterizes the asymptotic behavior of the variational Bayes approximation under model misspecification. I found none of the results particularly surprising since they are intuitive and expected, especially based on the results of [17] & [29]. (The proof also seems to be a straightforward extension.) I also find that the authors make too big of a deal about the model misspecificaiton error asymptotically dominating the variational approximation error, making this old observation look like a new contribution. (As an approximation becomes more accurate asymptotically, obviously the model misspecificaiton is going to dominate.) In particular, I find the section 2.2 verbose and repetitious in explaining the obvious intuition. The authors should be more explicit about the limitation of the result discussed in Section 2.3. I believe the local asymptotic normality assumption (Assumption 5) often fails under the type of models considered in Section 2.3 (e.g. hyper-parameters of the Gaussian process regression does not concentrate under the in-fill aymptotics). All that said, the result is a welcome addition to the literature on the theoretical properties of the variational Bayes approximation under model. I also appreciate that the authors kept the intuitions very clear (without making things unnecessarily complicated) and the paper is generally very easy to read. Minor comments: - Assumption 4 & 5 in the supplement are not just analogous but are essentially identical to Assumption 2 & 3. In this case, why not make the assumption for Section 2.3 more clear in the main manuscript? - Line 127, `tests $\phi_n$`: a `test` is undefined. I suppose it is a compactly supported smooth function, but is certainly not in the standard vocabulary of stats/ML audience. - Line 166, `limiting exact posterior`: this terminology threw me off and was very confusing to me because `limiting` and `exact` are contradictory. I suggest to call it just `limiting posterior`. - Line 166, `\theta` vs `\tilde{\theta}`: does the parameter with tilde play a different role? If not, it is just confusing. - Line 172, `\mathcal{Q}^d`: is the same as `\mathcal{Q}`? I suppose it is meant to emphasize the dependency on $d$, but the dependency was always there and the sudden change of notation is just confusing. - Line 289, `simulation corroborates... the limiting VB posterior coincide with the limiting exact posterior.`: I don't think this claim is true. Just looking at RMSE does *not* establish that that the two distributions (VB and MCMC) are close. Response to author feedbacks: Straitening out the main contributions and clarifying the limitations will certainly make the paper more worthwhile to the readers. With a successful revision, the paper will deserve the score of 7 (though there is no 2nd round review unfortunately).

Reviewer 2



The paper is clearly written and very well presented. Although inherently technical, the results are explained both precisely and in plain language, with proof sketches to convey intuitive understanding. This paper is a great model of clear communication of technical results. The results are novel to my knowledge and well situated in terms of previous literature. I found no obvious technical errors, although I wasn't able to closely check the proofs in the supplement. My impression is that the results themselves don't involve significant new technical ideas and are more or less straightforward extensions of previous theorems. Nonetheless, actually doing this technical work is a valuable contribution. My main concerns about significance, which largely apply to Bernstein-von Mises theorems more generally, is that by focusing on the asymptotic regime the work assumes away essentially all of the practically relevant structure in Bayesian inference problems. Behind all the technical machinery, the intuition behind these proofs (which, to its credit, the paper does a good job of conveying) is that for identifiable models in the iid asymptotic regime, the likelihood dominates the prior, and the posterior concentrates at a normal shrinking to a point mass, so we can ignore the prior and we can mostly ignore posterior uncertainty. But if you really believe you're in this regime, why not save yourself the trouble of VB and just fit an MLE? The argument that the MLE minimizes KL between the true data distribution and a misspecified model is so trivial that it's more of an observation (that the non-constant part of KL is just the expected model log likelihood) than an argument. This work dresses up that argument with substantially more mathematical machinery, but not (as far as I can tell) much more insight. It tells us that if you run VB in the setting where there is no uncertainty to quantify, it preserves the properties of a point estimate. This is well and good -- it's always possible that something could have gone wrong, and there's some pedantic value in checking that it doesn't -- but it's also kind of not the point of VB. Practical Bayesian inference involves quantifying uncertainty; without that, why are we here? We only get to do so much with our wild and precious lives, and it's not my place to question the authors' choices, but I can't help but view this as something of an example of math for math's sake with limited takeaways for the broader field. All that said, theoretical papers are in scope for NeurIPS, and this one is well done within (as far as I'm qualified to judge) the standards of the community.

Reviewer 3



Update After Rebuttal ---------------------- After reading rebuttal and other reviews, I continue to argue for acceptance and leave my original score (good paper, accept) unchanged. I thank the authors for willingness to discuss issues like local optima in the VB result in a revision, and also for willingness to describe simulations more carefully and share complete simulation code. I was glad to see comments from other reviewers about relevance of the LAN assumption or the Bernstein-von-Mises approach, and I hope a revision addresses these issues in more depth (as the rebuttal hints). In particular, I'd encourage a thoughtful response to the question R2 raises: "... if you really believe you're in this regime, why not save yourself the trouble of VB and just fit an MLE?". This which isn't really addressed in the rebuttal, and I think it's important to both raise and answer the point in the paper. Review Summary -------------- I appreciate the paper's focus on determining what happens to the optimal approximate posterior in the inevitable case that the model is "wrong", and thus the contribution of providing theorems to guide our understanding of how tractable approximations like VB behave in the asymptotic limit will be of interest at the conference. I wish the paper had a bit more to say about how local optima fit into this story (at least acknowledging the practical problems) and I'd like to see more details about the simulations, but overall this seems like interesting work and would lead to productive discussions as an accepted poster. Technical Comments ------------------ ## Comment on local optima? One inevitable issue with the practical outputs of variational Bayes iterative optimization is that we almost surely return a local optima rather than a global one (the variational ELBO is usually non-convex for any model of interest, such as the LDA topic model in the examples). While the theorems rather nicely govern the behavior of the global optima, I'm not sure they can say anything about local optima. Given this, I'm a bit surprised that the Simulations in Sec. 3 seem to gloss over the practical issues of local optima in VB as well as the practical mixing issues of MCMC on real finite datasets. For example, I'd be very reluctant to say that HMC has ever really converged to an "exact posterior" for a model like LDA, especially given only the results of one chain (often mixing issues aren't apparent until one chain does much better than 50 others). Perhaps with simple 15-dimensional observations it's possible to avoid serious problems (but still, if you look closely the curves don't match perfectly, so maybe there is a complication here). Would be interesting to see results from a much larger topic model fit (perhaps with several hundred observations), where local optima might play a larger role. ## Comments on rates of convergence? For practitioners that will never have access to "infinite" data, I wonder if the analysis here provides insight about when to expect that the convergence conditions are "close enough" to being satisfied (e.g. as argued in the Fig 2 plots, where 20000 examples is a proxy for "infinite" data). ## Tail condition assumption: How strong is it? When providing the key assumptions required for Thm. 1, while I agree it's likely true that many common priors satisfy the required tail condition -- second derivatives of log p(\theta) do not grow faster than exp(\theta^2) -- I wonder if there are any known (or easy-to-construct) counterexamples? It might help to give a bit more intuition for what kind of smoothness this implies. Comments on Simulations ----------------------- * Code to perform the simulation study is missing (seems that only the stan model specification is provided, not the code to produce the plots). Sharing this would be crucial to help readers reproduce results and understand their practical import. * How are the errorbars/intervals in Fig 2b calculated? Is this showing the full spread of the KL across topics, as well as average (mean) across topics? * Likewise, how are the errorbars in the remainder of Fig. 2 calculated? Should we be bothered that they do not seem to go to zero with more data (N)? * I might recommend that each figure show something like a 10x higher maximum N values than currently used. Doesn't seem you're showing real convergence (e.g. the intervals in Fig 2c at N=20000 don't quite line up yet). Presentation Comments --------------------- I found Sec. 1, especially the Main Idea, quite easy to follow. Well done. Fig. 1 does a nice job illustrating the main idea, with perhaps the exception that visually, the change in distribution over \theta from observing one to infinite data is larger than the whole space of possible factorized distributions Q.

[Author Response · NeurIPS 2019]

**Paper 7343 | Variational Bayes under Model Misspecification**

We thank the reviewers for their positive and constructive comments. All reviewers agree that characterizing variational Bayes under model misspecification is an interesting addition to the theory of variational Bayes literature. We are glad that the reviewers also appreciate the clear and intuitive explanations of technical results in this work, which could serve pedagogical purposes to the community. Below we respond to the main comments.

▌ **R1 finds the presentation in Section 2.2 and Assumptions 4 & 5 in Section 2.3 repetitive.**

Thank you for pointing it out. We will tighten up the explanation to make Theorem 2 clearer and more intuitive. We will also move Assumptions 4 & 5 into the main text to make Section 2.3 clearer.

▌ **R1 points out that the LAN assumption might not be satisfied in nonparametric models.**

Thank you for pointing it out. There are a few nonparametric models that have been shown to satisfy the LAN assumption, including generalized linear mixed models (Hall et al., 2011), stochastic block models (Bickel et al., 2013), and mixture models (Westling & McCormick, 2015). In Section 2.4, we apply Theorems 1 and 2 to these specific models and characterize their VB posteriors under different forms of model misspecification.

That said, we agree that the Bernstein-von Mises phenomenon does not hold in many nonparametric models with infinite dimensional parameters (Freedman, 1999). In these models, there is no posterior contraction or in-fill asymptotics for either the exact posterior or the VB posterior. We will clarify this limitation in the paper.

▌ **R2 is concerned about the practical relevance of Bernstein-von Mises type results.**

Thank you for pointing it out. We take the asymptotics perspective as a first step to understand the theoretical properties of VB posteriors and VB posterior predictive distributions. We were motivated by the empirical observation that variational Bayes predicts comparably with MCMC methods in large datasets. The results in this paper around the VB posterior predictive under model misspecification offers an explanation of this phenomenon. That said, we understand that Bernstein-von Mises type results can appear limited when the optimization complications of VB come in. We leave to future work the characterizations of how variational Bayes behaves in finite samples and how optimization complications affect the VB posterior.

▌ **R3 asks about how local optima in ELBO optimization fits into this story.**

We agree with R3 that local optima is a real practical issue in VB. We will add a discussion about local optima in the paper. The results in this work assume that the ELBO optimization returns a global optima. These results provide the possibility for local optima to share these properties, though further research is needed to understand the properties of local optima. For particular models like stochastic block models, Zhang and Zhou (2017) shows that global optima of the ELBO can be reached under weak conditions of optimization initialization. We believe that combining this work with optimization guarantees could lead to a fruitful further characterization of variational Bayes.

▌ **R3 are interested in seeing more details about the simulations, in particular the code, HMC mixing issues, VB optimization issues, error bars, and higher maximum N values.**

Thank you for the questions. We will include the figure-generation scripts in addition to current Stan code in the final version. Regarding practical HMC mixing and VB optimization issues, we follow the protocol as implemented in Stan. For HMC, we run four parallel chains and use 10,000 burn-in samples, and determine mixing using the R-hat convergence diagnostic (R-hat<1.01). For variational Bayes, we run optimization until convergence (i.e. a local optimum). We cannot confirm if the local optimal we reached is global. Further, we conduct multiple parallel runs under each simulation setup and report the mean and the standard deviation of "RMSE" or "Mean KL". The error bars in Figure 2 are the standard deviation across different runs of the same simulation. The simulation in the paper was conducted on a fairly small model, so the practical complications of HMC and VB did not appear detrimental. We agree that these practicalities are important, we will clarify these complications in the paper. We will also include higher maximum $N$ values to approximate the infinite data limit closer.

▌ **R3 asks about how large a dataset should be to approximate the infinite data limit.**

Thank you for the interesting question. When a model has more parameters or its convergence rate ($\delta_n$ in the LAN assumption) is lower, we should require a larger dataset to approximate the infinite data limit.

▌ **R3 asks about an example prior that fails the tail condition.**

Extreme value distributions like Gumbel distribution can fail this tail condition.

[Meta-Review · NeurIPS 2019]

The reviewers agree that this is a solid piece of work, and is of interest to the NeurIPS community. Please incorporate the changes mentioned in your rebuttal in the final submission.